# Error Amplification Limits ANN-to-SNN Conversion in Continuous Control

Zijie Xu [1 2]  Zihan Huang [2]  Yiting Dong [2]  Kang Chen [2]  Wenxuan Liu [2]  Zhaofei Yu [1 2]

## Abstract

Spiking Neural Networks (SNNs) can achieve competitive performance by converting already existing well-trained Artificial Neural Networks (ANNs), avoiding further costly training. This property is particularly attractive in Reinforcement Learning (RL), where training through environment interaction is expensive and potentially unsafe. However, existing conversion methods perform poorly in continuous control, where suitable baselines are largely absent. We identify error amplification as the key cause: small action approximation errors become temporally correlated across decision steps, inducing cumulative state distribution shift and severe performance degradation. To address this issue, we propose Cross-Step Residual Potential Initialization (CRPI), a lightweight gradient-free mechanism that carries over residual membrane potentials across decision steps to suppress temporally correlated errors. Experiments on continuous control benchmarks with both vector and visual observations demonstrate that CRPI can be integrated into existing conversion pipelines and substantially recovers lost performance. Our results highlight continuous control as a critical and challenging benchmark for ANN-to-SNN conversion, where small errors can be strongly amplified and impact performance. Code is available at https://github.com/xuzijie32/ANN2SNN-CRPI.

## 1. Introduction

Spiking Neural Networks (SNNs) (Maass, 1997; Gerstner et al., 2014) communicate through discrete spikes rather than continuous activations, enabling event-driven computa-

[1]Institute for Artificial Intelligence, Peking University, Beijing, China [2]Beijing Key Laboratory of Brain-inspired Spiking Large Models, School of Computer Science, Peking University, Beijing, China. Correspondence to: Wenxuan Liu, Zhaofei Yu <liuwx66@pku.edu.cn, yuzf12@pku.edu.cn>.

*Proceedings of the 43rd International Conference on Machine Learning*, Seoul, South Korea. PMLR 306, 2026. Copyright 2026 by the author(s).

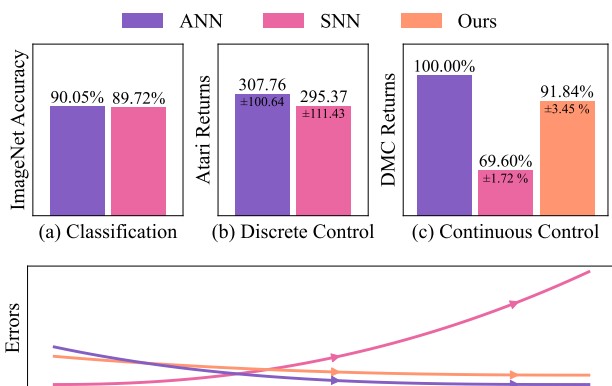

*Figure 1.* Challenges of ANN-to-SNN conversion across different task categories. (a) Classification accuracy on ImageNet (Huang et al., 2025). (b) Average returns in discrete control tasks on Atari (Patel et al., 2019). (c) Relative returns in continuous control tasks, averaged over six environments from the DeepMind Control Suite. Additional results in the experimental section confirm that the performance degradation is consistent across tasks. (d) Illustration of error accumulation and amplification, where trajectories generated by converted SNNs progressively diverge from those of the original ANN policies.

tion and substantially reducing energy consumption when deployed on neuromorphic hardware (Merolla et al., 2014; Davies et al., 2018; DeBole et al., 2019). These properties make SNNs particularly attractive for Reinforcement Learning (RL) on resource-constrained edge devices such as drones, wearables, and Internet-of-Things (IoT) sensors, where power efficiency is a critical concern.

ANN-to-SNN conversion constructs SNNs by transferring pretrained ANN weights and replacing nonlinear activations with spiking neurons, allowing SNNs to inherit strong ANN performance without additional training (Cao et al., 2015; Han et al., 2020; Li et al., 2021; Deng & Gu, 2021; Bu et al., 2022a; 2025). This gradient-free paradigm is especially valuable in RL, where learning an agent typically requires extensive environment interaction that is costly, time-consuming, and potentially unsafe (Jiang et al., 2023; Padalkar et al., 2024; Tang et al., 2025; Jayant & Bhatnagar, 2022). By reusing pretrained ANN policies, ANN-to-SNN conversion allows energy-efficient SNN agents to be deployed without extensive environment interaction.

Despite these advantages, the study of ANN-to-SNN conversion in RL remains limited in scope. Existing work has primarily focused on discrete control settings (Patel et al., 2019; Tan et al., 2021; Kumar et al., 2025; Feng et al., 2024), while ANN-to-SNN conversion in continuous action spaces remains largely unexplored, despite its central role in real-world robotics and embodied AI systems (Kober et al., 2013; Gu et al., 2017; Brunke et al., 2022). Figure 1(a)–(c) compares the performance of ANN-to-SNN conversion across classification, discrete control, and continuous control tasks. While existing conversion methods achieve competitive performance in classification and discrete control, they suffer substantially larger performance degradation in continuous control. This gap arises from the requirement for precise, high-dimensional vector-valued actions in continuous control, in contrast to the categorical outputs in classification and discrete control tasks, making continuous control considerably more sensitive to conversion errors.

To understand this phenomenon, we conduct a detailed analysis of conversion errors in continuous control. We find that: (i) performance degradation in converted SNNs is primarily driven by deviations in induced state trajectories rather than instantaneous action errors; (ii) these state deviations grow progressively over decision steps along a trajectory; and (iii) action approximation errors exhibit positive temporal correlation across consecutive decision steps, amplifying even small conversion errors. As illustrated in Figure 1(d), trajectories generated by converted SNNs gradually diverge from those of the optimal ANN policies, whereas ANN policies themselves do not exhibit such progressive drift.

Motivated by this analysis, we propose Cross-Step Residual Potential Initialization (CRPI), a simple yet effective mechanism to mitigate error amplification in ANN-to-SNN conversion for RL. CRPI carries over residual membrane potentials across consecutive decision steps to initialize neuron states, suppressing temporally correlated action errors and stabilizing the resulting state trajectories, as illustrated in Figure 1(d). Notably, CRPI requires no additional training and can be seamlessly integrated into existing gradient-free ANN-to-SNN conversion pipelines.

We evaluate CRPI on a range of continuous control benchmarks, including vector-based tasks from MuJoCo (Todorov et al., 2012) and vision-based environments from the DeepMind Control (DMC) Suite (Tunyasuvunakool et al., 2020). CRPI consistently improves the performance of multiple state-of-the-art ANN-to-SNN conversion methods and outperforms directly trained SNNs in challenging vision-based continuous control tasks. Our results highlight continuous control as a challenging benchmark for ANN-to-SNN conversion, where conversion errors can be strongly amplified and significantly impact long-horizon performance.

## 2. Related Works

### 2.1. ANN–SNN Conversion

ANN-to-SNN conversion typically maps ReLU activations in ANNs to the firing rates of Integrate-and-Fire neurons by accumulating spikes over time (Cao et al., 2015). However, the bounded firing rates in SNNs introduce significant errors, which is often mitigated through techniques such as weight normalization (Rueckauer et al., 2017) and threshold balancing (Han et al., 2020). Temporal discretization introduces additional quantization errors, which have been addressed by methods like quantizing the source ANN activations (Bu et al., 2023; Hu et al., 2023), using two-stage inference (Hao et al., 2023a), improving membrane potential initialization (Hao et al., 2023b), and extending neuron models with signed spikes (Wang et al., 2022a; Li et al., 2022) or multiple thresholds (Huang et al., 2024). Other encoding schemes such as time-to-first-spike coding (Rueckauer & Liu, 2018; Zhang et al., 2019; Stanojevic et al., 2023), phase coding (Kim et al., 2018; Wang et al., 2022b), burst coding (Park et al., 2019; Li & Zeng, 2022; Wang et al., 2025), and differential coding (Huang et al., 2025), have also been explored to enhance both efficiency and expressiveness. Recent works have further extended conversion methods by allowing for approximations of general nonlinear layers (Oh & Lee, 2024; Jiang et al., 2024; Huang et al., 2024) and enabling conversion of Transformer architectures to SNNs (Wang et al., 2023; You et al., 2024).

### 2.2. SNNs for Reinforcement Learning

Early works on SNNs for RL primarily relied on biologically inspired local learning rules, particularly reward-modulated spike-timing-dependent plasticity (R-STDP) and its variants (Florian, 2007; Frémaux & Gerstner, 2016; Gerstner et al., 2018; Frémaux et al., 2013; Yang et al., 2024). Later research introduced gradient-based optimization methods, such as spatio-temporal backpropagation (STBP) for deep spiking Q-networks (Wu et al., 2018; Liu et al., 2022; Chen et al., 2022; Qin et al., 2022; Sun et al., 2022) and e-prop for policy gradient methods (Bellec et al., 2020). Qin et al. 2025 further introduces gated recurrent mechanisms and demonstrates strong performance on partially observable tasks. In continuous control, the hybrid actor-critic framework has been widely adopted, where a spiking actor network is co-trained with an ANN-based critic network (Xu et al., 2025b; Tang et al., 2020; 2021; Zhang et al., 2022; Chen et al., 2024; Ding et al., 2022).

### 2.3. ANN–SNN Conversion in Reinforcement Learning

ANN-to-SNN conversion in RL has also been explored in several studies. These works mainly focus on converting Deep Q-Networks (DQNs) (Mnih, 2013; Mnih et al., 2015)

into spiking policies for Atari games (Patel et al., 2019; Tan et al., 2021), as well as deploying converted agents in real-world robotic tasks, such as ball catching (Feng et al., 2024) and path planning (Kumar et al., 2025). These studies report competitive performance, improved energy efficiency, and enhanced robustness of SNN-based agents. However, existing works have been limited to discrete control tasks, and ANN-to-SNN conversion in continuous control remains largely unexplored. This work shows that directly applying existing conversion techniques to continuous control leads to greater performance degradation, which forms the primary motivation for our work.

# 3. Preliminaries

## 3.1. Spiking Neural Networks

SNNs process information via discrete spike events and temporal membrane dynamics. For an Integrate-and-Fire (IF) neuron in layer $l$ at discrete time step $t$, the neuronal dynamics are given by

$$\mathbf{I}^l[t] = \mathbf{W}^l \mathbf{x}^{l-1}[t] + \mathbf{b}^l, \tag{1}$$

$$\mathbf{m}^l[t] = \mathbf{v}^l[t-1] + \mathbf{I}^l[t], \tag{2}$$

$$\mathbf{o}^l[t] = H\left(\mathbf{m}^l[t] - \boldsymbol{\theta}^l\right), \tag{3}$$

$$\mathbf{x}^l[t] = \boldsymbol{\theta}^l \odot \mathbf{o}^l[t], \tag{4}$$

$$\mathbf{v}^l[t] = \mathbf{m}^l[t] - \mathbf{x}^l[t], \tag{5}$$

where $\mathbf{x}^l[t]$ denotes the post-synaptic potential, $\mathbf{m}^l[t]$ and $\mathbf{v}^l[t]$ are the pre-reset and post-reset membrane potentials respectively, $\mathbf{o}^l[t]$ is the binary spike output, and $\boldsymbol{\theta}^l$ is the firing threshold. The operator $H(\cdot)$ denotes the Heaviside step function, and $\odot$ indicates element-wise multiplication.

## 3.2. ANN-to-SNN Conversion

ANN-to-SNN conversion leverages the correspondence between ReLU activations in ANNs and averaged firing responses in rate-coded SNNs. In a standard feedforward ANN, the output of layer $l$ is computed as

$$\mathbf{z}^l = \text{ReLU}\left(\mathbf{W}^l_{\text{ANN}} \mathbf{z}^{l-1} + \mathbf{b}^l_{\text{ANN}}\right). \tag{6}$$

Starting from the discrete-time dynamics of IF neurons, the membrane potential update can be written as

$$\mathbf{v}^l[t] = \mathbf{v}^l[t-1] + \mathbf{W}^l \mathbf{x}^{l-1}[t] + \mathbf{b}^l - \mathbf{x}^l[t]. \tag{7}$$

Averaging this equation over time steps $t = 1$ to $T$ yields

$$\frac{1}{T}\sum_{t=1}^{T}\mathbf{x}^l[t] = \mathbf{W}^l \frac{1}{T}\sum_{t=1}^{T}\mathbf{x}^{l-1}[t] + \mathbf{b}^l + \frac{\mathbf{v}^l[0] - \mathbf{v}^l[T]}{T}. \tag{8}$$

Under the standard assumption that the membrane potential of IF neurons remains bounded by the firing thresh-

old, i.e., $\mathbf{v}^l[t] \in [0, \boldsymbol{\theta}^l]$, the residual term $\frac{\mathbf{v}^l[0] - \mathbf{v}^l[T]}{T}$ vanishes as $T$ increases. By setting $\mathbf{W}^l = \mathbf{W}^l_{\text{ANN}}$ and $\mathbf{b}^l = \mathbf{b}^l_{\text{ANN}}$, and identifying ANN activations with the average post-synaptic potential $\mathbf{z}^{l-1} = \frac{1}{T}\sum_{t=1}^{T}\mathbf{x}^{l-1}[t]$, the time-averaged SNN response $\frac{1}{T}\sum_{t=1}^{T}\mathbf{x}^l[t]$ converges to the corresponding ANN activation $\mathbf{z}^l$.

## 3.3. Reinforcement Learning

RL studies the problem of an agent interacting with an environment, which is commonly formalized as a Markov Decision Process (MDP). At decision step $k$, the agent observes the environment state $\mathbf{s}_k \in \mathcal{S}$ and selects an action $\mathbf{a}_k \in \mathcal{A}$ according to a policy $\pi : \mathcal{S} \to \mathcal{A}$. The environment then transitions to a new state $\mathbf{s}_{k+1}$ and provides a reward $r_k = r(\mathbf{s}_k, \mathbf{a}_k)$. The objective of the agent is to maximize the expected cumulative return $R = \mathbb{E}\sum_k r_k$.

A key property of the MDP formulation is that the environment dynamics and reward depend only on the current state and action, rather than the full history of past interactions. Accordingly, at each decision step, the policy computes an action solely based on the current observation. In standard ANN-to-SNN conversion for RL, this is typically enforced by executing the SNN for a fixed internal simulation horizon of $T$ time steps at each decision step, and initializing all neuronal states at the beginning of the next decision step. As a result, no internal neuronal states or membrane potentials are preserved across consecutive decision steps.

# 4. Analyzing the Conversion Errors

This section investigates error propagation in ANN-to-SNN conversion for continuous control and identifies a phenomenon of error amplification. Section 4.1 decomposes the performance degradation of converted SNN policies into instantaneous action errors and the resulting state distribution shift, showing that the latter dominates the return loss. Section 4.2 demonstrates that small approximation errors are amplified across decision steps, leading to great state distribution shift. Section 4.3 identifies positive temporal correlations in action errors across consecutive decisions as the underlying cause of this amplification.

## 4.1. State-Dominated Performance Degradation

Given that ANN-to-SNN conversion exhibits greater performance degradation in continuous control than in widely-studied image classification tasks, we pose a central question: **Is this error solely due to the conversion process, or is it also amplified by the dynamics of RL environments?**

We begin by formalizing the expected return of a policy $\pi$:

$$R^\pi = \mathbb{E}_{s \sim P_\pi(s), a = \pi(s)}\left[r(s, a)\right], \tag{9}$$

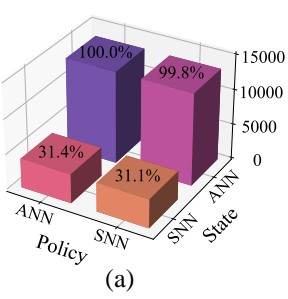
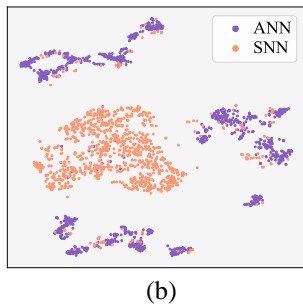

(a)        (b)

*Figure 2.* Analysis of performance degradation in ANN-to-SNN conversion in the HalfCheetah-v4 environment. The ANN policy is trained with TD3 for 3 million environment steps and converted using IF neurons with 8 simulation steps. (a) Expected returns under different combinations of policies and state distributions. (b) t-SNE visualization of state trajectories induced by ANN and converted SNN policies, revealing significant distribution divergence.

where $P_\pi(s)$ denotes the marginal state distribution induced by executing policy $\pi$ in the environment. Accordingly, the expected returns of the original ANN policy and its converted SNN counterpart are given by

$$R^{\text{ANN}} = \mathbb{E}_{s\sim P_\pi^{\text{ANN}}(s), a=\pi^{\text{ANN}}(s)}[r(s,a)], \quad (10)$$

$$R^{\text{SNN}} = \mathbb{E}_{s\sim P_\pi^{\text{SNN}}(s), a=\pi^{\text{SNN}}(s)}[r(s,a)]. \quad (11)$$

The discrepancy between $R^{\text{ANN}}$ and $R^{\text{SNN}}$ arises from two sources: (i) divergence in the state visitation distributions, i.e., $P_\pi^{\text{ANN}}$ versus $P_\pi^{\text{SNN}}$, and (ii) differences in action selection induced by the converted policy, i.e., $\pi^{\text{ANN}}$ versus $\pi^{\text{SNN}}$. To disentangle these effects, we define two auxiliary returns:

$$R^{\text{SNN}|\text{ANN}} = \mathbb{E}_{s\sim P_\pi^{\text{ANN}}(s),\, a=\pi^{\text{SNN}}(s)}[r(s,a)], \quad (12)$$

$$R^{\text{ANN}|\text{SNN}} = \mathbb{E}_{s\sim P_\pi^{\text{SNN}}(s),\, a=\pi^{\text{ANN}}(s)}[r(s,a)]. \quad (13)$$

The **SNN-action-only return** $R^{\text{SNN}|\text{ANN}}$ evaluates the effect of replacing the ANN policy with the converted SNN policy while keeping the ANN-induced state distribution fixed. Conversely, the **SNN-state-only return** $R^{\text{ANN}|\text{SNN}}$ isolates the impact of state distribution shift induced by the converted SNN while preserving the original ANN policy.

Figure 2(a) reports the expected returns of the ANN, SNN, SNN-action-only, and SNN-state-only settings. Replacing $\pi^{\text{ANN}}$ with $\pi^{\text{SNN}}$ while maintaining the ANN-induced state distribution results in only negligible performance degradation (less than $0.5\%$). In contrast, executing either policy under the SNN-induced state distribution leads to a substantial reduction in return. Comprehensive experiments results in Appendix B.1 also demonstrates same pattern exists across diverse RL environments and SNN settings. This indicates that the performance degradation is overwhelmingly dominated by deviations in the induced state trajectories

rather than instantaneous action mismatches. Furthermore, Fig. 2(b) visualizes the divergence between state trajectories generated by ANN and SNN policies. Despite their close per-step action outputs, the resulting trajectories diverge greatly, highlighting the sensitivity of continuous control systems to small perturbations.

### 4.2. Error Accumulation and Amplification

Having identified state distribution shift as the primary source of performance degradation, a natural question arises: **how does this shift evolve along a trajectory? Is it uniformly distributed, or does it grow over time?**

In a Markov Decision Process, each action affects the subsequent state, which in turn influences all future transitions. Consequently, small action errors introduced by ANN-to-SNN conversion can propagate across decision steps, causing progressive state deviations and accumulating performance loss.

Figure 3 (a) illustrates how state trajectories induced by ANN and converted SNN policies diverge over time. The discrepancy is small at early stages but grows progressively as interaction proceeds, demonstrating that state deviations accumulate and are amplified by the environments. Figure 3 (b) then shows the corresponding impact on returns. While ANN and SNN policies achieve similar rewards at the start of an episode, the gap steadily widens over the decision horizon, reflecting the cumulative effect of state divergence on long-horizon performance.

### 4.3. Positive Cross-Step Correlation

The analysis in Section 4.2 shows that conversion errors accumulate and amplify over decision steps. A natural question arises: **why do these errors persist instead of being corrected by subsequent actions?**

We analyze the correlation of action approximation errors across consecutive steps. At step $k$ with state $s_k$, let the actions produced by the ANN and the converted SNN be $a_k^{\text{ANN}}$ and $a_k^{\text{SNN}}$, and define the instantaneous action error as $\delta a_k = a_k^{\text{SNN}} - a_k^{\text{ANN}}$. Executing these actions leads to next states $s_{k+1}^{\text{ANN}}$ and $s_{k+1}^{\text{SNN}}$. To separate the effect of policy response from state shift, we define a counterfactual action $a_{k+1}^{\text{cf}} = \pi^{\text{ANN}}(s_{k+1}^{\text{SNN}})$, which applies the ANN policy to the SNN-induced next state.

Using these definitions, we compute three cosine similarity metrics that characterize cross-step error propagation:

**ANN Correction** measures whether the ANN policy compensates for the previous-step action error under the shifted state:

$$\text{ANN Correction} = \cos\left(\delta a_k,\ a_{k+1}^{\text{cf}} - a_{k+1}^{\text{ANN}}\right). \quad (14)$$

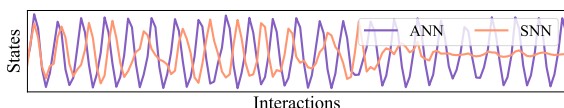

*(a)* State evolution over decision steps.

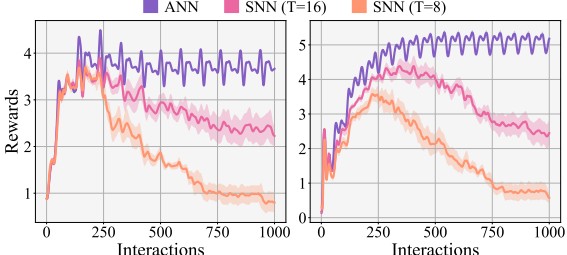

*(b)* Average reward per decision step.

*Figure 3.* (a) One-dimensional visualization of state evolution over decision steps for ANN and converted SNN policies in the HalfCheetah-v4 environment, obtained by projecting paired trajectories onto the first principal component via PCA. (b) Average reward per decision step for ANN and converted SNN policies in Hopper-v4 (left) and Walker2d-v4 (right). Shaded regions denote half a standard deviation. All curves are uniformly smoothed for clarity. The ANN was trained with TD3 for 3 million environment steps, and the SNN uses IF neurons.

*Table 1.* Cosine similarity of action errors across consecutive decision steps in different environments. The ANN is trained with TD3 for 3 million environment steps and the SNN uses IF neurons with 16 simulation steps.

| Environment | ANN Correction | SNN Consistency | SNN Drift |
|---|---|---|---|
| Ant-v4 | $-0.188 \pm 0.004$ | $0.276 \pm 0.018$ | $0.030 \pm 0.019$ |
| HalfCheetah-v4 | $-0.070 \pm 0.023$ | $0.101 \pm 0.033$ | $0.043 \pm 0.043$ |
| Hopper-v4 | $-0.256 \pm 0.005$ | $0.481 \pm 0.016$ | $0.129 \pm 0.013$ |
| Walker2d-v4 | $-0.185 \pm 0.009$ | $0.462 \pm 0.015$ | $0.180 \pm 0.020$ |

**SNN Consistency** captures whether the SNN exhibits similar action deviations across consecutive steps under the same shifted state:

$$\text{SNN Consistency} = \cos\left(\delta a_k, \, a_{k+1}^{\text{SNN}} - a_{k+1}^{\text{cf}}\right). \quad (15)$$

**SNN Drift** directly quantifies the temporal correlation of action errors between ANN and SNN trajectories:

$$\text{SNN Drift} = \cos\left(\delta a_k, \, a_{k+1}^{\text{SNN}} - a_{k+1}^{\text{ANN}}\right). \quad (16)$$

Intuitively, negative ANN Correction indicates that the ANN policy actively compensates for previous errors, whereas positive SNN Consistency shows that errors persist across steps. Table 1 confirms this: ANN policies display negative temporal correlations, demonstrating inherent error-correcting behavior, while converted SNNs exhibit positive correlations (SNN Consistency) and consistent drift (SNN Drift). These results suggest that temporally correlated action errors are the key mechanism behind error accumulation and amplification in ANN-to-SNN conversion.

## 5. Reducing the Compounding Errors

The analysis in Section 4 shows that the performance degradation of ANN-to-SNN conversion in continuous control is primarily driven by positively correlated action approximation errors across consecutive decision steps. Once such temporal correlations arise, even small conversion errors are repeatedly reinforced by the environment dynamics, inducing progressive state drift and ultimately leading to amplified performance loss. Our objective is therefore to explicitly suppress this cross-step error correlation.

### 5.1. Deriving the Methods

Motivated by the prior analyses of rate-based ANN-to-SNN conversion (Bu et al., 2022b), we assume that the dominant source of action approximation error arises from residual membrane potentials at the end of each decision step $k$ in ANN-to-SNN conversion:

$$\varepsilon_k^l = \frac{\mathbf{v}_k^l[T] - \mathbf{v}_k^l[0]}{T}. \quad (17)$$

We use the temporal correlation of residual membrane potential errors as a tractable proxy for action-level error correlation.

Empirical results in Section 4.3 indicate that these residual errors exhibit positive temporal correlation, i.e., $\mathbb{E}\left[\cos\left(\varepsilon_{k+1}^l, \varepsilon_k^l\right)\right] > 0$, which directly leads to systematic error accumulation across decision steps. Rather than minimizing the magnitude of $\varepsilon_k^l$ independently at each step, we instead aim to suppress its temporal correlation. Formally, our goal is to enforce

$$\mathbb{E}\left[\cos\left(\tilde{\varepsilon}_{k+1}^l, \varepsilon_k^l\right)\right] \leq 0, \quad (18)$$

where $\tilde{\varepsilon}_{k+1}^l$ denotes the modified residual error after applying a cross-step correction.

To this end, we consider a first-order approximation of residual error dynamics across consecutive decision steps:

$$\tilde{\varepsilon}_{k+1}^l = \varepsilon_{k+1}^l - \alpha \, \varepsilon_k^l, \quad (19)$$

where $\alpha > 0$ captures the empirically observed positive alignment between successive residual errors (as demonstrated in Table 1). Increasing $\alpha$ reduces the expected cosine similarity in Eq. (18) and can drive it below zero, thereby suppressing error accumulation.

Substituting the definition of $\varepsilon_k^l$ into Eq. (19) yields

$$\tilde{\varepsilon}_{k+1}^l = \frac{\mathbf{v}_{k+1}^l[T] - \mathbf{v}_{k+1}^l[0]}{T} - \alpha \frac{\mathbf{v}_k^l[T] - \mathbf{v}_k^l[0]}{T} \quad (20)$$

$$= \frac{\mathbf{v}_{k+1}^l[T] - \left(\mathbf{v}_{k+1}^l[0] - \alpha \left(\mathbf{v}_k^l[T] - \mathbf{v}_k^l[0]\right)\right)}{T}. \quad (21)$$

**Algorithm 1** Inference with CRPI

---

Initialize membrane potentials for all layers $\mathbf{v}_0^l[0] \leftarrow \frac{1}{2}\theta^l$
Observe initial environment state $s_0$
Run SNN for $T$ steps and execute action $a_0 = \pi^{\mathrm{SNN}}(s_0)$
**for** $k = 1$ to $K$ **do**
    Observe next state $s_k$
    Compute residual potential from previous step:

$$\Delta\mathbf{v}_k^l \leftarrow \mathbf{v}_{k-1}^l[T] - \mathbf{v}_{k-1}^l[0]$$

    Clip residual to ensure valid firing range:

$$\Delta\mathbf{v}_k^l \leftarrow \max\big(\Delta\mathbf{v}_k^l, \; -\Sigma_{t=1}^{T}\mathbf{x}_{k-1}^l[t]\big)$$

    Initialize membrane potentials with cross-step residual:

$$\mathbf{v}_k^l[0] \leftarrow \mathrm{clip}\big(\tfrac{1}{2}\theta^l + \alpha\Delta\mathbf{v}_k^l, 0, \; \theta^l\big)$$

    Run SNN for $T$ steps and execute $a_k = \pi^{\mathrm{SNN}}(s_k)$
**end for**

---

Under the standard assumption that the final membrane potential is approximately uniformly distributed in $(0, \theta^l)$ (Bu et al., 2022a) and weakly dependent on its initialization (supported by empirical evidence in Appendix B.2), the expected residual error can be reduced by adjusting the initial membrane potential as

$$\mathbf{v}_{k+1}^l[0] \leftarrow \mathbf{v}_{k+1}^l[0] + \alpha\left(\mathbf{v}_k^l[T] - \mathbf{v}_k^l[0]\right). \tag{22}$$

### 5.2. Cross-Step Residual Potential Initialization

We refer to the membrane potential initialization mechanism of Equation (22) as Cross-Step Residual Potential Initialization (CRPI). In practice, Equation (17) assumes non-negative activations induced by ReLU. Therefore, we clip $\mathbf{v}_k^l[T] - \mathbf{v}_0^l[T]$ in Equation (22) to a minimum of $-\sum_{t=1}^{T}\mathbf{x}_k^l[t]$ to remain consistent with Equation (6). Moreover, to prevent excessively large residuals from inducing abnormally high initial membrane potentials that may cause persistent bursting across subsequent decision steps, we further clip $\mathbf{v}_{k+1}^l[0]$ to the valid membrane potential range. The complete procedure is summarized in Algorithm 1.

CRPI introduces no additional training or architectural modification and operates solely through membrane potential initialization. It is lightweight and can be readily integrated with existing ANN-to-SNN conversion techniques such as normalization, quantization, and neuron model extensions.

## 6. Experiments

### 6.1. Experimental Setup

**Environments.** We evaluate CRPI on a diverse set of continuous control benchmarks with both vector-based

and vision-based observations. For vector-based control, we consider four standard MuJoCo environments (Todorov et al., 2012; Todorov, 2014) from OpenAI Gymnasium (Brockman, 2016; Towers et al., 2024): Ant (Schulman, 2015), HalfCheetah (Wawrzyński, 2009), Hopper (Erez et al., 2012), and Walker2d. For vision-based control, we evaluate six tasks from the DeepMind Control (DMC) Suite (Tunyasuvunakool et al., 2020): Cartpole_Swingup, Finger_Spin, Reacher_Easy, Cheetah_Run, Acrobot_Swingup, and Quadruped_Walk.

**RL Algorithms.** For vector-based environments, we use ANN policies pre-trained with three sample-efficient off-policy algorithms: Deep Deterministic Policy Gradient (DDPG) (Lillicrap, 2015), Twin Delayed DDPG (TD3) (Fujimoto et al., 2018), and Soft Actor-Critic (SAC) (Haarnoja et al., 2018a;b). Each policy is trained for 3 million environment steps. For vision-based environments, we adopt ANN policies trained with Data-Regularized Q-v2 (DrQ-v2) (Yarats et al., 2021b;a) for 1 million environment steps.

**ANN-to-SNN Conversion Methods.** We integrate CRPI with multiple ANN-to-SNN conversion techniques. As a baseline, we apply CRPI to standard IF neurons. To assess compatibility with more expressive neuron models, we further combine CRPI with Signed Neuron Models (SNM) (Wang et al., 2022a), Multi-Threshold Neurons (MT) (Huang et al., 2024), and Differential Coding (DC) (Huang et al., 2025). All MT neurons use four firing thresholds. To avoid additional hyperparameter tuning, the firing threshold $\theta$ is set to the maximum ReLU activation channel-wise.

**Evaluation Protocol.** All results are averaged over five random seeds. For each seed, we evaluate the policy over ten rollout episodes of up to $1{,}000$ interaction steps (terminated earlier if the episode ends), yielding a total of $50{,}000$ environment steps per method. The hyperparameter $\alpha$ is selected via a coarse grid search over $\{0, 0.1, 0.2, \ldots, 0.9, 1.0\}$ and is fixed across all seeds.

### 6.2. Reducing Error Correlation

To evaluate the effectiveness of CRPI in mitigating the temporally correlated conversion errors identified in Section 4, we first examine how CRPI influences cross-step error correlation. Specifically, we analyze the cosine similarity of residual membrane potential errors across adjacent decision steps, together with the *SNN Consistency* and *SNN Drift* metrics introduced in Section 4.3, which quantify temporal correlation at the action level.

Figure 4 shows that the temporal correlation of residual membrane potential errors decreases monotonically as $\alpha$ increases, indicating that CRPI effectively suppresses cross-step error correlation. Importantly, this reduction at the

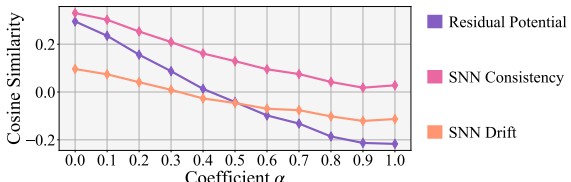

*Figure 4.* Cosine similarity of residual membrane potential and action errors across consecutive decision steps under different values of $\alpha$. Results are obtained on MuJoCo environments using TD3 and IF neurons with 16 simulation steps.

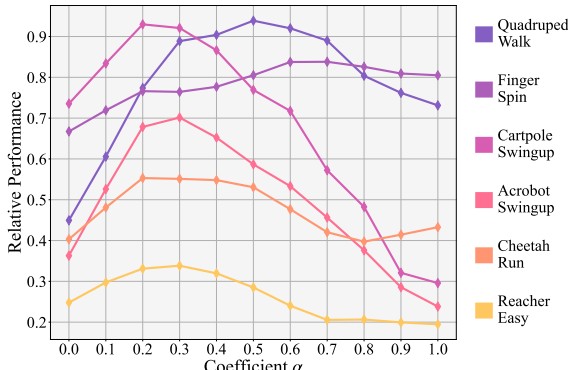

*Figure 5.* Relative performance on DeepMind Control tasks under different correlation parameter $\alpha$, using IF neurons with 32 simulation steps. Performance is normalized by the corresponding ANN return. Curves are uniformly smoothed for visualization.

membrane level consistently propagates to the action level: both the SNN Consistency and SNN Drift metrics are substantially reduced as $\alpha$ increases. These results provide direct empirical evidence that CRPI decorrelates conversion-induced errors across decision steps, addressing the root cause of error amplification identified in Section 4.3.

### 6.3. Enhancing Performance

We next examine how this reduction in temporal error correlation translates into policy performance. Figure 5 illustrates the relationship between the correlation coefficient $\alpha$ and relative performance (the converted SNN's performance normalized by the ANN baseline) across tasks. Starting from $\alpha = 0$ (standard ANN-to-SNN conversion), increasing $\alpha$ leads to a steady improvement in performance, demonstrating that incorporating cross-step residual information effectively mitigates temporally correlated conversion errors. However, when $\alpha$ becomes too large, performance begins to degrade. That is because excessive values of $\alpha$ overcompensate residual errors from previous steps, resulting in unstable membrane potential initialization and greater output error. This behavior reveals a clear trade-off between error decorrelation and overcorrection.

Overall, these results show that CRPI prevents small approximation errors from being repeatedly reinforced by closed-loop system dynamics. By suppressing temporal error correlation, CRPI alleviates the error amplification phenomenon analyzed in Section 4.2 and stabilizes long-horizon behavior in continuous control tasks, which we identified as the dominant factor governing performance degradation in Section 4.1.

### 6.4. Comprehensive Benchmarks

We conduct comprehensive evaluations on both MuJoCo and DeepMind Control Suite (DMC) benchmarks to assess the effectiveness of CRPI under a wide range of settings, including different observation modalities, neuron models, SNN simulation steps, environments, and underlying RL algorithms. Tables 2 and 3 report the performance of the converted SNN policies on MuJoCo and DMC tasks. For

each configuration, we report the Average Performance Ratio (APR), defined as the mean ratio of SNN performance to the corresponding ANN performance, expressed in percentage and averaged across all evaluated environments.

Across all evaluated configurations, CRPI consistently improves the performance ratio compared to the corresponding baseline ANN-to-SNN conversion methods[1]. These improvements are observed consistently across different neuron models, simulation lengths, and RL algorithms, indicating that CRPI is robust to both architectural and algorithmic variations. Detailed results for individual MuJoCo environments are provided in Appendix B.3, where CRPI demonstrates consistent performance gains across all tasks.

Table 3 additionally includes comparisons with state-of-the-art directly trained SNNs on visual-based DMC tasks, including approaches incorporating a world model (Hafner et al., 2019). Specifically, we compare against vanilla Leaky Integrate-and-Fire (LIF) neurons, the two-component spiking neuron model (TC-LIF) (Zhang et al., 2024), and the spiking world model (Spiking-WM) (Sun et al., 2025). CRPI consistently outperforms these directly trained SNN approaches, highlighting the advantage of leveraging well-trained ANN policies while effectively mitigating error accumulation and amplification during the conversion process.

### 6.5. Energy Efficiency

We further analyze the inference-time energy consumption of the converted SNN models. Following the widely adopted

---

[1]In several cases, the converted SNNs slightly outperform their ANN counterparts. This phenomenon may be attributed to the inherent stochasticity of RL environments, where the event-driven dynamics and robustness of SNNs can better tolerate noise (Ding et al., 2025). Similar observations have also been reported in prior studies on ANN-to-SNN conversion for discrete control tasks (Patel et al., 2019; Tan et al., 2021).

*Table 2.* Performance comparison of the average performance ratio of ANN-to-SNN conversion on MuJoCo continuous control tasks.

| NEURON | T | CONVERTING DDPG | | | CONVERTING TD3 | | | CONVERTING SAC | | |
|---|---|---|---|---|---|---|---|---|---|---|
| | | ORIGINAL | OURS | Δ | ORIGINAL | OURS | Δ | ORIGINAL | OURS | Δ |
| IF | 8 | 77.96% | 87.78% | +9.82% | 64.71% | 72.26% | +7.55% | 70.25% | 75.38% | +5.13% |
| | 16 | 82.57% | 95.41% | +12.84% | 79.11% | 85.10% | +5.99% | 88.80% | 93.09% | +4.29% |
| | 32 | 94.24% | 105.07% | +10.83% | 89.68% | 93.52% | +3.84% | 97.98% | 99.70% | +1.71% |
| SNM | 8 | 65.69% | 76.72% | +11.04% | 82.97% | 89.24% | +6.27% | 79.56% | 87.89% | +8.34% |
| | 16 | 95.14% | 102.06% | +6.92% | 96.70% | 98.18% | +1.48% | 92.13% | 98.09% | +5.96% |
| MT | 2 | 68.74% | 86.74% | +18.00% | 76.29% | 80.18% | +3.89% | 84.02% | 90.06% | +6.03% |
| | 4 | 85.68% | 102.22% | +16.54% | 97.93% | 98.09% | +0.16% | 95.29% | 97.44% | +2.15% |
| DC | 2 | 74.32% | 83.97% | +9.65% | 89.39% | 93.65% | +4.27% | 86.58% | 92.93% | +6.35% |
| | 4 | 101.65% | 104.81% | +3.16% | 99.30% | 99.80% | +0.50% | 96.85% | 101.72% | +4.88% |

*Table 3.* Performance comparison on DeepMind Control Suite tasks with visual observations, where ± denotes half a standard deviation.

| MODULE | T | CRPI | ACROBOT SWINGUP | CARTPOLE SWINGUP | CHEETAH RUN | FINGER SPIN | QUADRUPED WALK | REACHER EASY | APR |
|---|---|---|---|---|---|---|---|---|---|
| ANN | – | – | 225±17 | 880±0 | 733±4 | 976±1 | 751±9 | 929±19 | 100.00% |
| LIF | 8 | – | 104.0 | 774.1 | 515.5 | 416.2 | 259.7 | 403.4 | 54.19% |
| TC-LIF | 8 | – | 106.5 | 667.7 | 517.8 | 657.8 | 302.2 | 613.2 | 61.25% |
| SPIKING-WM | 8 | – | 113.7 | 791.0 | 577.2 | 682.0 | 350.7 | 701.3 | 68.54% |
| IF | 32 | × | 57±9 | 563±99 | 240±34 | 606±39 | 295±25 | 197±59 | 40.77% |
| | | √ | 167±21 | 823±8 | 411±48 | 831±11 | 752±22 | 334±58 | 74.19% |
| | 64 | × | 93±12 | 843±16 | 494±42 | 896±12 | 643±26 | 331±24 | 69.60% |
| | | √ | 223±24 | 867±1 | 642±20 | 940±5 | 774±21 | 617±29 | 91.84% |
| SNM | 8 | × | 216±19 | 853±4 | 612±30 | 913±9 | 772±24 | 551±33 | 88.72% |
| | | √ | 248±25 | 853±4 | 637±23 | 933±5 | 774±18 | 787±35 | 96.27% |
| | 16 | × | 217±21 | 879±0 | 714±6 | 963±6 | 758±22 | 907±17 | 98.50% |
| | | √ | 237±26 | 879±0 | 728±9 | 972±1 | 781±6 | 933±19 | 101.41% |
| MT | 2 | × | 210±17 | 878±0 | 558±43 | 952±5 | 752±29 | 947±17 | 94.85% |
| | | √ | 244±28 | 878±0 | 701±7 | 954±1 | 783±8 | 947±17 | 101.30% |
| DC | 2 | × | 215±20 | 877±0 | 620±5 | 969±1 | 749±20 | 899±17 | 95.97% |
| | | √ | 248±29 | 877±0 | 660±25 | 969±1 | 768±32 | 939±20 | 100.43% |

*Table 4.* Average inference-time energy consumption of ANNs and converted SNNs in DeepMind Control suite.

| | ANN | IF (T=32) | MT (T=2) |
|---|---|---|---|
| FLOPS | $4.53 \times 10^7$ | – | – |
| SOPS | – | $2.12 \times 10^8$ | $2.71 \times 10^7$ |
| CONSUMPTIONS | 566.84 $\mu$J | 16.35 $\mu$J | 2.09 $\mu$J |

estimation framework in (Merolla et al., 2014), we approximate energy expenditure by assigning 12.5 pJ per floating-point operation (FLOP) and 77 fJ per synaptic operation (SOP) (Qiao et al., 2015; Hu et al., 2021).

As shown in Table 4, ANN baselines incur substantially higher energy consumption per inference compared to their converted SNN counterparts. Despite requiring multiple simulation steps, SNNs achieve significant energy savings due to their sparse, event-driven computation. With ad-

vanced multi-threshold neurons, both the number of operations and the energy consumption are further reduced.

It is worth noting that the membrane potential initialization mechanism in CRPI introduces negligible energy overhead (less than 1%) compared to vanilla SNNs. More detailed results and discussions regarding energy efficiency can be found in Appendix B.4.

These results highlight the energy efficiency of SNNs and further support the suitability of CRPI-converted SNNs for low-power and resource-constrained deployment scenarios.

## 7. Conclusion

This work presents a systematic study of ANN-to-SNN conversion in continuous control and identifies a fundamental limitation absent in classification and discrete control tasks: small approximation errors become temporally corre-

lated through long-horizon interactions, inducing progressive state drift and severe performance degradation. To mitigate this effect, we propose Cross-Step Residual Potential Initialization, a gradient-free inference mechanism that suppresses cross-step error correlation. CRPI is compatible with diverse neuron models and conversion schemes, and consistently improves performance on MuJoCo and DeepMind Control benchmarks while preserving energy efficiency. Our findings position continuous control as a critical benchmark for evaluating ANN-to-SNN conversion, where even minor approximation errors can be greatly amplified and result in severe performance degradation. Future work will focus on developing adaptive tuning schemes for the correlation parameter $\alpha$ and extending the CRPI framework to broader sequential generation tasks vulnerable to error accumulation, such as language modeling and diffusion processes.

## Acknowledgements

This work is supported by the National Natural Science Foundation of China (62422601, U24B20140, 62506011), Beijing Municipal Science and Technology Program (Z251100008125052), and Qiyuan Innovative Talent Program.

## Impact Statement

This work aims to advance ANN-to-SNN conversion for continuous control by analyzing the sources of conversion errors and proposing an inference-time method to mitigate temporally correlated error accumulation. By improving the stability and effectiveness of converted SNN policies in long-horizon decision-making tasks, this work may facilitate the deployment of energy-efficient spiking models in resource-constrained control systems such as robotics and embedded platforms. We **do not** anticipate significant ethical or societal risks beyond those commonly associated with control and reinforcement learning applications.

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

# A. Additional Experiments Details

## A.1. Reinforcement Learning Environments

We evaluate the proposed method on a diverse set of continuous control benchmarks covering both vector-based and vision-based observations. Specifically, we consider standard MuJoCo (Todorov et al., 2012; Todorov, 2014) tasks from OpenAI Gymnasium (Brockman, 2016; Towers et al., 2024) and visual control tasks from the DeepMind Control Suite (DMC) (Tunyasuvunakool et al., 2020). These environments are widely used for benchmarking reinforcement learning algorithms.

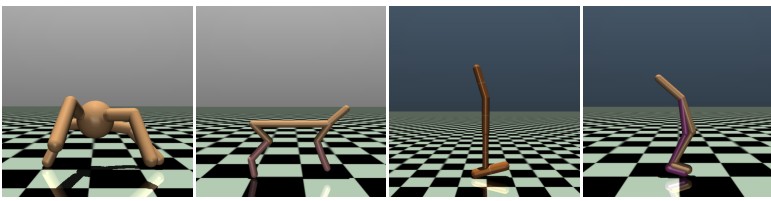

*Figure 6.* Representative MuJoCo continuous control tasks used in our experiments. From left to right: Ant-v4, HalfCheetah-v4, Hopper-v4, and Walker2d-v4.

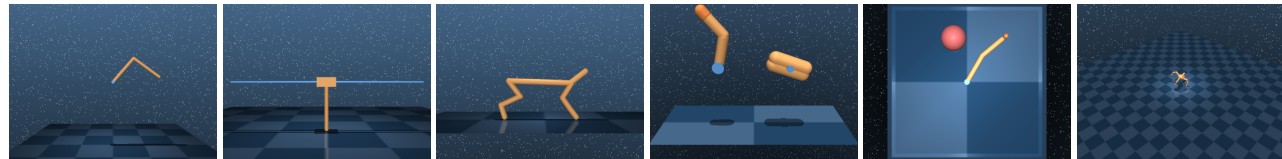

*Figure 7.* Representative DeepMind Control Suite tasks used in our experiments. From left to right: acrobot_swingup, cartpole_swingup, cheetah_run, finger_spin, reacher_easy, and quadruped_walk.

As shown in Figure 6, the MuJoCo environments includes `Ant` (Schulman, 2015), `HalfCheetah` (Wawrzyński, 2009), `Hopper` (Erez et al., 2012), and `Walker2d`. Besides, the DMC Suite includes `cartpole_swingup`, `finger_spin`, `reacher_easy`, `cheetah_run`, `acrobot_swingup`, and `quadruped_walk`, demonstrated in Figure 7.

*Table 5.* State and action space dimensions for MuJoCo environments.

| ENVIRONMENT | STATE DIMENSION | ACTION DIMENSION |
|---|---|---|
| ANT-V4 | 27 | 8 |
| HALFCHEETAH-V4 | 17 | 6 |
| HOPPER-V4 | 11 | 3 |
| WALKER2D-V4 | 17 | 6 |

Tables 5 and 6 summarize the dimensionalities of state and action spaces for all evaluated environments. MuJoCo tasks use low-dimensional vector states, while DMC tasks rely on pixel observations; in all cases, the action space is continuous.

All environments are used with their default parameters provided by the respective simulators. Rewards are not rescaled or normalized during either training or evaluation. For evaluation, each episode is capped at a maximum horizon of 1000 environment interactions, unless terminated earlier by environment-specific conditions. These settings are kept consistent across ANN baselines and converted SNN policies to ensure fair comparison.

## A.2. Reinforcement Learning Algorithms

For vector-based continuous control tasks in MuJoCo, we employ three widely used off-policy actor–critic algorithms: Deep Deterministic Policy Gradient (DDPG) (Lillicrap, 2015), Twin Delayed DDPG (TD3) (Fujimoto et al., 2018), and Soft Actor-Critic (SAC) (Haarnoja et al., 2018a;b). All three methods learn a deterministic (DDPG, TD3) or stochastic (SAC) policy together with one or more Q-function critics. The actor networks in all methods are implemented as multilayer

*Table 6.* Observation and action space specifications for DeepMind Control Suite environments.

| DOMAIN NAME | TASK NAME | OBSERVATION | ACTION DIMENSION |
|---|---|---|---|
| ACROBOT | SWINGUP | $84 \times 84 \times 3$ | 1 |
| CARTPOLE | SWINGUP | $84 \times 84 \times 3$ | 1 |
| FINGER | SPIN | $84 \times 84 \times 3$ | 2 |
| REACHER | EASY | $84 \times 84 \times 3$ | 2 |
| CHEETATH | RUN | $84 \times 84 \times 3$ | 6 |
| QUADRUPED | WALK | $84 \times 84 \times 3$ | 12 |

perceptrons (MLPs) with two hidden layers and ReLU activations. DDPG has 400 hidden units in layer 1 and 300 hidden units in layer 2. TD3 and SAC have 256 units in both hidden layers.

For image-based continuous control tasks in the DeepMind Control Suite, we adopt Data-Regularized Q-v2 (DrQ-v2) (Yarats et al., 2021b;a), a sample-efficient off-policy algorithm designed for high-dimensional visual observations. DrQ-v2 employs a convolutional encoder followed by an actor-critic architecture. Specifically, the visual encoder consists of four convolutional layers with $3 \times 3$ kernels and 32 channels, followed by a fully connected layer that produces a compact latent representation of 50 dimensions. Both the actor and critic networks operate on this latent feature and are implemented as two-layer MLPs with 1024 hidden units.

All RL agents are trained using standard hyperparameters and default environment settings. After training, the actor networks are converted to SNNs using different ANN-to-SNN conversion methods. During evaluation, only the converted SNN policies interact with the environment, and no further learning or fine-tuning is performed.

### A.3. ANN-to-SNN Conversion Approaches

Our experiments evaluate three baseline methods: Signed Neuron with Memory (SNM) (Wang et al., 2022a), Multi Threshold (MT) Neuron (Huang et al., 2024), and Differential coding (DC) based neuron(Huang et al., 2025).

#### A.3.1. SNM NEURON DYNAMICS

SNM neuron can be regarded as an IF neuron with negative threshold and more strict spike emission condition on negative threshold in SNNs, let $\boldsymbol{m}^l(t)$ and $\boldsymbol{v}^l(t)$ denote the membrane potential of neurons in the $l$-th layer before and after firing spikes at time-step $t$, the neural dynamic can be formulated as follows:

$$\boldsymbol{m}^l(t) = \boldsymbol{v}^l(t-1) + \boldsymbol{W}^l \boldsymbol{x}^{l-1}(t), \tag{23}$$

$$\boldsymbol{s}^l(t) = H(\boldsymbol{m}^l(t) - \theta^l) - H(-\boldsymbol{m}^l(t) + \theta^l) \cdot H(\boldsymbol{c}^l(t) + \theta^l), \tag{24}$$

$$\boldsymbol{x}^l(t) = \theta^l \boldsymbol{s}^l(t), \tag{25}$$

$$\boldsymbol{v}^l(t) = \boldsymbol{m}^l(t) - \boldsymbol{x}^l(t). \tag{26}$$

$$\boldsymbol{c}^l[t] = \boldsymbol{c}^l[t-1] + \boldsymbol{x}^l[t], \tag{27}$$

where $H$ is the Heaviside step function and $\theta^l$ is the neuron threshold in layer $l$. $\boldsymbol{s}^l(t)$ is the output spike of layer $l$. $\boldsymbol{x}^l(t)$ is the postsynaptic potential and theoretical output of layer $l$. $\boldsymbol{c}^l[t-1]$ represents an auxiliary cumulative variable used to support the ReLU-like behavior.

#### A.3.2. MT NEURON DYNAMICS

The MT neuron is characterized by several parameters, including the base threshold $\theta$, and a total of $2n$ thresholds, with $n$ positive and $n$ negative thresholds. The threshold values of the MT neuron are indexed by $i$, where $\lambda_i^l$ represents the $i$-th threshold value in the layer $l$:

$$\lambda_1^l = \theta^l, \lambda_2^l = \frac{\theta^l}{2}, ..., \lambda_n^l = \frac{\theta^l}{2^{n-1}},$$

$$\lambda_{n+1}^l = -\theta^l, \lambda_{n+2}^l = -\frac{\theta^l}{2}, ..., \lambda_{2n}^l = -\frac{\theta^l}{2^{n-1}}. \tag{28}$$

Let variables $\boldsymbol{I}^l[t]$, $\boldsymbol{W}^l$, $\boldsymbol{s}^l_i[t]$, $\boldsymbol{x}^l[t]$, $\boldsymbol{m}^l[t]$, and $\boldsymbol{v}^l[t]$ represent the input current, weight, the output spike of the $i$-th threshold, the total output signal, and the membrane potential before and after spikes in the $l$-th layer at the time-step $t$. It defines $\frac{4}{3}m^l[t] = (-1)^S 2^E(1+M)$ with 1 sign bit ($S$), 8 exponent bits ($E$), and 23 mantissa bits ($M$). Since the median of $\frac{1}{2^{k-1}}$ and $\frac{1}{2^k}$ is $\frac{3}{4}\frac{1}{2^{k-1}}$, we can easily select the correct threshold index $i$ using $E$ and $S$ of $\frac{4}{3}m^l[t]$. The dynamics of the MT neurons are described by the following equations:

$$\boldsymbol{m}^l[t] = \boldsymbol{v}^l[t-1] + \boldsymbol{I}^l[t] = \boldsymbol{v}^l[t-1] + \boldsymbol{x}^{l-1}[t], \tag{29}$$

$$\boldsymbol{s}^l_i[t] = \text{MTH-R}_{\theta,n}(\boldsymbol{m}^l[t], i) \tag{30}$$

$$\boldsymbol{x}^l[t] = \sum_i \boldsymbol{s}^l_i[t]\boldsymbol{W}^l\lambda^l_i, \tag{31}$$

$$\boldsymbol{v}^l[t] = \boldsymbol{m}^l[t] - \boldsymbol{x}^l[t], \tag{32}$$

$$\text{MTH-R}_{\theta,n}(\boldsymbol{m}^l[t], i) = \begin{cases} 1, & \text{if} \begin{cases} i < n, & S = 0 \text{ and } i-1 = -E, \\ i \geq n, & S = 1 \text{ and} \\ & i-n-1 = \max(-E, -E_2) \end{cases} \\ 0, & \text{otherwise.} \end{cases} \tag{33}$$

$$\tag{34}$$

$$\boldsymbol{c}^l[t] = \boldsymbol{c}^l[t-1] + \boldsymbol{x}^l[t], \tag{35}$$

where $\boldsymbol{c}^l[t-1] = (-1)^{S_2}2^{E_2}(1+M_2)$ represents an auxiliary cumulative variable used to support the ReLU-like behavior.

### A.3.3. DC based neuron dynamics

In rate coding, the output of the previous layer, $\boldsymbol{x}^{l-1}[t]$, is directly used as the input current for the current layer $\boldsymbol{I}^l[t] = \boldsymbol{x}^{l-1}[t]$. In differential coding, the input current $\boldsymbol{I}^l[t]$ can be adjusted as shown in Equation (36), which converts any spiking neuron into a differential spiking neuron:

$$\boldsymbol{I}^l[t] = \boldsymbol{m}^l_r[t] + \boldsymbol{x}^{l-1}[t], \tag{36}$$

$$\boldsymbol{m}^l_r[t+1] = \boldsymbol{m}^l_r[t] + \frac{\boldsymbol{x}^{l-1}[t]}{t} - \frac{\boldsymbol{x}^l[t]}{t}, \tag{37}$$

where $\boldsymbol{m}^l_r[0]$ is $\boldsymbol{b}^{l-1}$ if the previous layer has bias else 0. This work employs differential coding methods based on MT neurons.

For linear layers, including fully connected and convolutional layers that can be represented by Equation (38),

$$\boldsymbol{x}^l = \boldsymbol{W}^l\boldsymbol{x}^{l-1} + \boldsymbol{b}^l, \tag{38}$$

where $\boldsymbol{W}^l$ and $\boldsymbol{b}^l$ is the weight and bias of layer $l$. Under differential coding in SNNs, this is equivalent to eliminating the bias term $\boldsymbol{b}^l$ and initializing the membrane potential of the subsequent layer with the bias value as Equation (39):

$$\boldsymbol{x}^l = \boldsymbol{W}^l\boldsymbol{x}^{l-1}. \tag{39}$$

## B. Additional Experiments Results

### B.1. Additional Results on Reward Decomposition

This section provides additional empirical results for the reward decomposition analysis introduced in Section 4.1. The expected return is decomposed into four counterfactual settings: (i) the original ANN policy and state trajectory $R^{\text{ANN}}$, (ii) the fully converted SNN policy and state trajectory $R^{\text{SNN}}$, (iii) ANN policy evaluated on SNN-induced state trajectories $R^{\text{ANN|SNN}}$, and (iv) SNN policy evaluated on ANN-induced state trajectories $R^{\text{SNN|ANN}}$.

Table 7 reports detailed results for Integrate-and-Fire (IF) neurons converted from TD3 policies on MuJoCo environments under different SNN simulation time steps. Across all environments and time horizons, we observe a consistent pattern that replacing the policy alone while keeping ANN state trajectories results in only marginal performance degradation, whereas

*Table 7.* Reward decomposition results for ANN-to-SNN conversion using IF neurons. ANNs are trained with TD3 in MuJoCo environments for 3 million interactions.

| ENVIRONMENT | $T$ | $R^{\mathrm{ANN}}$ | $R^{\mathrm{SNN|ANN}}$ | $R^{\mathrm{ANN|SNN}}$ | $R^{\mathrm{SNN}}$ |
|---|---|---|---|---|---|
| ANT-V4 | 8 | 6505.26 | 6216.71 | 4051.77 | 3988.25 |
| | 16 | | 6393.15 | 6112.45 | 6008.63 |
| | 32 | | 6475.02 | 6263.13 | 6294.63 |
| HALFCHEETAH-V4 | 8 | 13193.35 | 13164.41 | 4146.40 | 4104.06 |
| | 16 | | 13179.90 | 6477.50 | 6445.50 |
| | 32 | | 13190.03 | 9720.97 | 9711.38 |
| HOPPER-V4 | 8 | 3594.20 | 3605.06 | 3519.23 | 3532.34 |
| | 16 | | 3602.29 | 3560.10 | 3572.05 |
| | 32 | | 3599.08 | 3575.31 | 3580.73 |
| WALKER2D-V4 | 8 | 4582.30 | 4620.69 | 3118.59 | 3122.68 |
| | 16 | | 4610.12 | 3471.64 | 3475.12 |
| | 32 | | 4598.57 | 4058.01 | 4065.86 |

replacing the state trajectory leads to substantial return loss, even when the ANN policy is used. This further confirms that state distribution shift is the dominant factor driving performance degradation in ANN-to-SNN conversion for continuous control.

These results align with the main findings in the paper and demonstrate that the state-dominated performance gap persists across different environments and SNN time resolutions.

### B.2. Distribution of the Final Membrane Potential

In Section 5.1, we derive the CRPI mechanism under the assumption that the final membrane potential is approximately uniformly distributed in $(0, \theta^l)$ (Bu et al., 2022a) and weakly dependent on its initialization. To empirically validate this assumption, we analyze the distribution of final membrane potentials for a converted TD3 agent (IF neuron, T=64) in the Hopper-v4 environment.

*Table 8.* Distribution of final membrane potentials under different initializations.

| INITIALIZATION | $(-\infty, 0]$ | $(0, 0.1\theta]$ | $(0.1\theta, 0.2\theta]$ | $(0.2\theta, 0.3\theta]$ | $(0.3\theta, 0.4\theta]$ | $(0.4\theta, 0.5\theta]$ | $(0.5\theta, 0.6\theta]$ | $(0.6\theta, 0.7\theta]$ | $(0.7\theta, 0.8\theta]$ | $(0.8\theta, 0.9\theta]$ | $(0.9\theta, \theta]$ | $(\theta, +\infty)$ |
|---|---|---|---|---|---|---|---|---|---|---|---|---|
| 0 | 85.3% | 1.4% | 1.5% | 1.5% | 1.5% | 1.5% | 1.5% | 1.4% | 1.5% | 1.4% | 1.4% | 0.2% |
| $0.25\theta$ | 85.2% | 1.5% | 1.5% | 1.5% | 1.5% | 1.5% | 1.4% | 1.4% | 1.5% | 1.4% | 1.4% | 0.2% |
| $0.5\theta$ | 85.2% | 1.5% | 1.5% | 1.5% | 1.5% | 1.5% | 1.4% | 1.4% | 1.4% | 1.4% | 1.4% | 0.2% |
| $0.75\theta$ | 85.1% | 1.5% | 1.5% | 1.5% | 1.5% | 1.5% | 1.5% | 1.5% | 1.4% | 1.4% | 1.4% | 0.2% |
| $\theta$ | 85.1% | 1.5% | 1.5% | 1.5% | 1.5% | 1.5% | 1.5% | 1.5% | 1.5% | 1.4% | 1.4% | 0.2% |

Table 8 shows that most neurons have negative potentials, where they are mostly inactive (and clipped in CRPI), and are thus irrelevant to the mechanism. For active neurons, membrane potentials are approximately uniformly distributed over $(0, \theta]$, with only a small fraction exceeding $\theta$ (and are also mostly clipped). Furthermore, this distribution remains nearly unchanged across initializations, indicating weak dependence on initialization.

### B.3. Detailed MuJoCo Results across RL Algorithms

This section provides detailed per-environment results for ANN-to-SNN conversion on MuJoCo benchmarks, complementing the aggregated results reported in the main paper. Due to space constraints, the main text only presents averaged performance metrics across environments. Here, we report the full breakdown across individual tasks.

Note that DDPG fails to converge reliably on the `Ant` environment, which is consistent with prior observations in the literature (Xu et al., 2025a). As a result, results for DDPG are reported only on the remaining three MuJoCo tasks. TD3 and SAC results include all four environments.

Across all settings, CRPI consistently improves conversion performance relative to standard initialization, with gains observed across environments, time steps, and neuron types. These detailed results further support the robustness and generality of the proposed method.

*Table 9.* Detailed ANN-to-SNN conversion results on MuJoCo environments using DDPG, where $\pm$ captures half a standard deviation.

| NEURON | TIME | CRPI | HALFCHEETAH | HOPPER | WALKER | APR |
|---|---|---|---|---|---|---|
| ANN | – | – | 9126±129 | 2703±121 | 1982±201 | 100.00% |
| IF | $T = 8$ | × | 4486±337 | 1097±54 | 2856±214 | 77.96% |
| | | √ | 4486±337 | 1410±87 | 3212±299 | 87.78% |
| | $T = 16$ | × | 5744±319 | 1689±86 | 2424±287 | 82.57% |
| | | √ | 5799±230 | 2021±63 | 2931±159 | 95.41% |
| | $T = 32$ | × | 6591±419 | 2311±120 | 2477±278 | 94.24% |
| | | √ | 7321±543 | 2815±116 | 2593±66 | 105.07% |
| SNM | $T = 8$ | × | 5600±434 | 1529±75 | 1568±263 | 65.69% |
| | | √ | 6725±650 | 1620±63 | 1913±259 | 76.72% |
| | $T = 16$ | × | 8086±490 | 2092±136 | 2367±178 | 95.14% |
| | | √ | 8500±435 | 2490±154 | 2397±329 | 102.06% |
| MT | $T = 2$ | × | 7100±605 | 1737±38 | 1272±164 | 68.74% |
| | | √ | 7615±556 | 1874±107 | 2129±336 | 86.74% |
| | $T = 4$ | × | 8445±350 | 2210±113 | 1640±140 | 85.68% |
| | | √ | 9040±356 | 2310±231 | 2420±219 | 102.22% |
| DC | $T = 2$ | × | 6926±493 | 1681±79 | 1682±262 | 74.32% |
| | | √ | 7485±518 | 1817±104 | 2035±142 | 83.97% |
| | $T = 4$ | × | 8644±222 | 2428±214 | 2387±68 | 101.65% |
| | | √ | 8818±405 | 2605±191 | 2407±189 | 104.81% |

## B.4. computational overhead of CRPI

During deployment, the parameter $\alpha$ in CRPI can be selected from the set 0, 0.125, 0.25, 0.375, 0.5, 0.625, 0.75, 0.875, 1, where each value can be expressed as a sum of powers of 1/2. The multiplication between $\alpha$ and the residual membrane potential can be efficiently implemented using at most three bitwise shifts and three floating-point accumulation operations (ACs). Other operations in CRPI (e.g., addition and clipping) do not involve multiplication. This design effectively eliminates multiplication operations during deployment, leading to minimal overhead.

Table 12 reports the average number of ACs introduced by CRPI compared to standard forward propagation. Across different neuron models, CRPI contributes less than 1% additional ACs, confirming its negligible overhead.

*Table 10.* Detailed ANN-to-SNN conversion results on MuJoCo environments using TD3, where $\pm$ captures half a standard deviation.

| NEURON | T | CRPI | ANT | HALFCHEETAH | HOPPER | WALKER | APR |
|---|---|---|---|---|---|---|---|
| ANN | – | – | 6505±127 | 13193±12 | 3594±1 | 4582±4 | 100.00% |
| IF | 8 | × | 3988±526 | 4104±522 | 3532±3 | 3123±275 | 64.71% |
| | | √ | 4261±236 | 4625±400 | 3560±2 | 4098±130 | 72.26% |
| | 16 | × | 6009±252 | 6445±592 | 3572±1 | 3475±324 | 79.11% |
| | | √ | 6009±252 | 7261±331 | 3576±1 | 4285±86 | 85.10% |
| | 32 | × | 6295±201 | 9711±273 | 3581±1 | 4066±210 | 89.68% |
| | | √ | 6666±48 | 9938±493 | 3586±1 | 4422±80 | 93.52% |
| SNM | 8 | × | 5025±358 | 10014±600 | 3358±105 | 3909±161 | 82.97% |
| | | √ | 5430±456 | 10014±600 | 3583±4 | 4486±70 | 89.24% |
| | 16 | × | 6138±242 | 12168±323 | 3592±1 | 4594±3 | 96.70% |
| | | √ | 6423±124 | 12365±259 | 3594±1 | 4595±3 | 98.18% |
| MT | 2 | × | 2637±325 | 10120±178 | 3364±92 | 4323±105 | 76.29% |
| | | √ | 2961±254 | 10967±237 | 3396±98 | 4473±78 | 80.18% |
| | 4 | × | 6359±160 | 12720±181 | 3592±1 | 4473±86 | 97.93% |
| | | √ | 6359±160 | 12720±181 | 3592±1 | 4501±87 | 98.09% |
| DC | 2 | × | 6014±145 | 11061±130 | 3329±98 | 4062±220 | 89.39% |
| | | √ | 6091±138 | 12043±166 | 3514±36 | 4212±153 | 93.65% |
| | 4 | × | 6408±287 | 13033±30 | 3593±1 | 4579±4 | 99.30% |
| | | √ | 6538±199 | 13033±30 | 3593±1 | 4579±4 | 99.80% |

*Table 11.* Detailed ANN-to-SNN conversion results on MuJoCo environments using SAC, where $\pm$ captures half a standard deviation.

| NEURON | T | CRPI | ANT | HALFCHEETAH | HOPPER | WALKER | APR |
|---|---|---|---|---|---|---|---|
| ANN | – | – | 6829±140 | 14967±22 | 3385±89 | 5030±70 | 100.00% |
| IF | 8 | × | 3258±237 | 7960±478 | 2966±116 | 4651±150 | 70.25% |
| | | √ | 4127±168 | 8043±318 | 3017±111 | 4941±87 | 75.38% |
| | 16 | × | 5123±284 | 11406±213 | 3555±7 | 4978±90 | 88.80% |
| | | √ | 5944±164 | 11617±239 | 3555±7 | 5165±21 | 93.09% |
| | 32 | × | 6715±98 | 13462±110 | 3535±30 | 4990±62 | 97.98% |
| | | √ | 6755±98 | 13570±88 | 3612±3 | 5156±15 | 99.70% |
| SNM | 8 | × | 5709±374 | 6584±538 | 3155±54 | 4901±81 | 79.56% |
| | | √ | 6539±37 | 7662±442 | 3493±65 | 5101±34 | 87.89% |
| | 16 | × | 6832±141 | 10727±494 | 3233±192 | 5096±36 | 92.13% |
| | | √ | 6966±9 | 12066±242 | 3660±30 | 5111±11 | 98.09% |
| MT | 2 | × | 6579±108 | 7908±766 | 3037±153 | 4889±147 | 84.02% |
| | | √ | 6628±90 | 9073±681 | 3476±98 | 5022±25 | 90.06% |
| | 4 | × | 6823±122 | 11850±308 | 3490±114 | 4979±80 | 95.29% |
| | | √ | 7003±11 | 11917±508 | 3588±103 | 5110±21 | 97.44% |
| DC | 2 | × | 5729±332 | 10123±359 | 3204±193 | 5037±35 | 86.58% |
| | | √ | 6375±143 | 10955±343 | 3518±94 | 5091±20 | 92.93% |
| | 4 | × | 6668±195 | 13780±302 | 3315±120 | 5016±82 | 96.85% |
| | | √ | 7020±12 | 14035±69 | 3672±22 | 5122±18 | 101.72% |

*Table 12.* Average computational overhead of CRPI on the DeepMind Control Suite.

| SPIKING NEURON | T | ACS IN CRPI | ACS IN FORWARD PROPAGATION | CRPI OVERHEAD |
|---|---|---|---|---|
| IF | 32 | $1.76 \times 10^5$ | $2.12 \times 10^8$ | 0.08% |
| MT | 2 | $1.88 \times 10^5$ | $2.71 \times 10^7$ | 0.69% |

