# OpenReview forum: "Error Amplification Limits ANN-to-SNN Conversion in Continuous Control"
_ICML.cc/2026/Conference — ICML 2026 regular_

### Official Review · Reviewer_66vP · 2026-03-04

**Soundness:** 4
**Presentation:** 4
**Significance:** 3
**Originality:** 3
**Overall Recommendation:** 6
**Confidence:** 3

**Summary:**

In this work, the authors introduce an ANN-to-SNN conversion in the context of continuouis control reinforcement Learning.Such setting has been largely overlooked in prior work.

The main contributions can be summarized as follows:
* The authors identify error amplification as the central difficulty. Indeed, small action approximation errors that are introduced by the conversion become temporally correlated across decisiion steps, which causes cumulative state distribution shift and performance degradation;
* The authors provide a systematic analysis decomposing the return gap into action-error and state-shift components. They introduce cosine similarity metrics to measure cross-step error correlation. They propose CRPI, a mechanism that carries over residual membrane potentials between decision steps to suppress temporal error correlation;
* Experiments are conducted in MuJoCo and deepming Control Suite tasks with multiple RL algorithms and neuronal models.

Overall, this manuscript proposes a solution for ANN-to-SNN conversion in continuous control reinforcement Learning.

**Compliance With Llm Reviewing Policy:**

Affirmed.

**Final Justification:**

All my concerns have been adequately addressed in the rebuttal. The authors provided convincing additional experiments and proposed to revise the manuscript accordingly. I am raising my score (presentation to 4 and overall to 6). I recommend the acceptance of this paper.

**Key Questions For Authors:**

Based on the previous weaknesses, here are my key questions for the authors:
* Can you provide empirical evidence for the uniformity assumption ?
* Have you considered experiments with per-layer alpha values rather than a single global scalar ?
* How does CRPI compare against light fine-tuning of the converted SNN ?
* Why is alpha restricted to non-negative values ?

Thanks.

**Limitations:**

Yes

**Strengths And Weaknesses:**

# Strengths
I would summarize the strengths of the work as follows:
* The problem identification is the paper strongest contribution in my opinion. Indeed, establishing that continuous control is qualitatively harder for ANN-to-SNN conversion than classification or discrete control, and carefully diagnosing why,is extremely valuable for the neuromorphic community;
* The auxiliary return decomposition presented in section 4.1, Eqs. 12-13, is very well designed. Disentangling policy-level action error from state distribution shift by cross-evualating policies on each other induced state distribution is an elegan experimental methodology;
* The counterfactual action framework of Section 4.1 is also an elegant analytical contribution;
* CRPI itself is simple, principled and practical. Indeed, no retraining nor major architectural changes are required. Moreover, the experimental coverage is broad;
* The organization of the paper is logical, which gives a nice structure, especially with the questions organizing the sections. The open source code shared with the reviewers is also appreciated.

# Weaknesses
However, I feel that the paper could be greatly improved by addressing the following criticisms:
* Variability reporting is systematically insufficient throughout the paper, which weakens the empirical claims. Indeed, for instance, Figure 1 (b) and (c) aggregate over multiple environments but show only single bars. Error bars or, better, per-environment scatter points overlaid on the bar would be much more informative. Another examples lies in Table 1, which reports only mean cosine similarities. However, since these are computed over trajectories and environment, standard deviation or confidence intervals are needed to assess reliability;
* The "training-free" framing is overstated. Indeed, Figure 5 clearly shows that the optimal values of alpha vary a lot across tasks (from 0.3 to 0.7 roughly). The authors use a coarse grid search over alpha, which requires rolling out the SNN in the environment multiple times. While it is true that no gradient computation is involved, finding a good alpha still requires non-trivial environment interaction. I would suggest to modify the introduction and abstract to reflect this more honestly. Moreover, if tuning is required anyway, why restrict alpha to a single global scalar? Per-layer alpha values could potentially improve performance at modest additional tuning cost;
* The paper shows that ANN policies exhibit negative temporal error correlation (self-correction) while converted SNN shows positive corrélations (error persistence), but the explanation for why this asymmetry arises is shallow from my understanding. This is Worth discussing more carefully. However, I agree that this comment might emerge from my own limited understanding of prior work, like Bu et al. (2022b, 2023, 2025) that is cited as the basis for the residual membrane potential error model, and I would appreciate if the authors could comment on what other sources of conversion error exist beyond residual membrane potentials and how those interact with the error amplification phenomenon described here;
* There is a lack of comparison against a natural and important Baseline: taking the converted SNN and continue the training with a small number of environment steps. Please include this in your study or discuss why it should not be included;
* The theorertical derivation of CRPI (5.1) relies on the assumption that the final memebrane potential is approximately uniformly distribution in [0, theta] and weakly dépendent on its initialization (Bu et al., 2022a). This is a critical assumption that is not empirically validated anywhere in the paper. Please include such analysis.

Below, the authors can found some minor, point-wise comments that could be easily implemented to improve the quality of the manuscript:
* Section 6.5 feels out of place in the main text, and the Energy estimates use standard formulas from prior work, while not interacting with CRPI at all. I would recommend Moving this section to the appendix for free space;
* Inconsistent spacing before punctuation in équations;
* A typo around line 308;
* "relative performance" in Section 6.3 is not defined near its first use.

# Overall assessment
While this contribution is timely, sound and well presented in the field of ANN-to-SNN conversion in reinforcement learning, the results critically lack a variability reporting that would help to assess the reliability of the method. However, the broad experimental validation and shared code is greatly appreciated. I would encourage the authors to review their manuscript to improve its significance. Beforehand, I personnally thank the authors that have read this review carefully.

---

> ### Author Rebuttal · Authors · 2026-03-30
>
> We sincerely appreciate your thorough review and the invaluable insights. We will strive to address your questions.
>
> > W1: Variability reporting
>
> Thank you for pointing this out. For Fig. 1(b), the mean ± std of ANNs and SNNs in discrete control are 307.06±100.64 and 295.37±111.43 (as reported in [1]). For Fig. 1(c), the average performance across environments for vanilla SNNs and SNNs with CRPI are 69.60%±1.72% and 91.84%±3.45% (mean ± std over environments). For Table 1, we additionally report the standard deviations of cosine similarities over 5 seeds:
>
> Table R1: Cosine similarities (mean ± std).
> |Environment|ANN Correction|SNN Consistency|SNN Drift|
> |-|:-:|:-:|:-:|
> |Ant-v4|-0.188±0.004|0.276±0.018|0.030±0.019|
> |HalfCheetah-v4|-0.070±0.023|0.101±0.033|0.043±0.043|
> |Hopper-v4|-0.256±0.005|0.481±0.016|0.129±0.013|
> |Walker2d-v4|-0.185±0.009|0.462±0.015|0.180±0.020|
>
> The variances in Fig. 1(c) and Table R1 are small relative to mean differences, supporting the robustness of our conclusions. We will revise the manuscript to include these variability statistics.
>
> > W2(a): The "training-free" framing
>
> Thank you for this suggestion. We will replace “training-free” with the more precise term “gradient-free”, and clarify in the manuscript that a small amount of environment interaction is still required.
>
> > W2(b) & Q2: Per-layer alpha
>
> Extending alpha from a global scalar to per-layer (or even per-channel) parameters could improve performance, as the current setting is a constrained special case.
>
> However, this increases the search space from 1D to multi-dimensional, leading to higher environment interaction costs. Since our primary goal is to identify key issues in ANN-to-SNN conversion for continuous control and provide a simple and effective solution, we adopt a global alpha to keep the method lightweight.
>
> We believe that fine-grained alpha is promising. Exploring efficient parameter search or adaptive schemes to support this setting is an important direction for future work, which we will discuss in the revision.
>
> > W3(a): Why SNNs show positive correlations
>
> Due to discretization, similar input states may produce identical spike outputs in SNNs, while their ANN activations remain close but distinct. Consequently, their conversion errors become positively correlated.
>
> In RL, state transitions are typically smooth, so consecutive states are highly similar. This leads to temporally correlated conversion errors when combined with discrete SNNs.
>
> > W3(b): Other sources of conversion error
>
> From Eqs. (6–8), assuming both ANN and SNN layers receive the same input from the previous layer, with $a^l=W^lz^{l-1}+b^l$, the conversion error is $a^l-ReLU(a^l)-\frac{v^l[T]-v^l[0]}{T}$.
>
> 1. When $a^l\ge0$, the error reduces to the residual membrane potential term.
>
> 2. When $a^l<0$, the error equals the number of emitted spikes (since $ReLU(a^l)=0$), and the residual potential alone is insufficient. To address this, we introduce a lower bound on the residual membrane potential (inverse of spike count) in the CRPI mechanism (as shown in Algorithm 1), ensuring the error is properly captured.
>
> >W4 & Q3: Comparison with light fine-tuning
>
> Thank you for this insightful suggestion. We consider two post-conversion fine-tuning strategies:
> 1. Continuing RL training;
> 2. Distillation from the ANN policy.
>
> To ensure a fair comparison, all methods use the same number of environment interactions as in the grid search of CRPI. Table R2 shows the average performance ratio (compared with ANNs) on MuJoCo tasks, using the IF neuron with the DDPG algorithm.
>
> Table R2: Performance comparison with fine-tuning baselines.
> |T|Conversion|+ Continue Training|+ Distillation|+ CRPI|
> |:-:|:-:|:-:|:-:|:-:|
> |8|77.96%|56.34%|72.30%|**87.78%**|
> |16|82.57%|57.00%|84.17%|**95.41%**|
> |32|94.24%|72.53%|94.79%|**105.07%**|
>
> Continuing RL training after conversion consistently degrades performance, which is because RL algorithms are sensitive to abrupt policy shifts introduced during conversion. Distillation provides moderate gains in some cases. In contrast, CRPI consistently achieves the best performance across all settings, demonstrating both effectiveness and stability.
>
> > W5 & Q1: On the assumption that "the final membrane potential is approximately uniformly distributed in (0, θ) and weakly dependent on its initialization"
>
> Due to space limits, please refer to our response to Reviewer snnE (Weakness 2).
>
> > W6-9: Writing suggestions
>
> Thank you for these helpful suggestions. We will revise the manuscript accordingly.
>
> > Q4: Why is alpha restricted to non-negative values?
>
> In principle, alpha can take any value. However, CRPI aims to reduce error correlations. A negative alpha introduces positive feedback, amplifying correlations and degrading performance.
>
> References:
>
> [1] Patel, D., et al. (2019). Improved robustness of reinforcement learning policies upon conversion to spiking neuronal network platforms applied to atari breakout game. Neural Networks.

---

> > ### Author Rebuttal · Reviewer_66vP · 2026-04-02
> >
> > Thanks for this rebuttal! I think that you addressed all my points.
> >
> > - "Gradient-free" is indeed a better term than "training-free" to describe the method.
> > - Your answer about alpha per layer is indeed valid. I see in other reviewers’ discussions that they also suggest making it learnable; this would be interesting future work.
> > - I’m very surprised by the results on distillation and fine-tuning: I think they should really be part of the main text.
> > - My questions were mostly for personal understanding, and you answered all of them - thanks.
> >
> > I’m changing my score. Please introduce the variability score in the text and in the figure.
> >
> > Best

---

> > > ### Author Response · Authors · 2026-04-02
> > >
> > > We are glad that our response has addressed your concerns. We will incorporate your suggested modifications into the final manuscript. Thank you very much for helping us significantly improve our paper!

---

### Official Review · Reviewer_UPqJ · 2026-03-11

**Soundness:** 4
**Presentation:** 3
**Significance:** 4
**Originality:** 4
**Overall Recommendation:** 5
**Confidence:** 5

**Summary:**

This paper studies performance degradation in ANN-to-SNN conversion for continuous control and attributes the issue to temporally correlated action errors across decision steps. To address this problem, the authors propose Cross-Step Residual Potential Initialization (CRPI), which carries residual membrane potentials from previous steps to reduce error amplification. The method is integrated into standard ANN-to-SNN conversion pipelines and evaluated on MuJoCo and DeepMind Control benchmarks with DDPG, TD3, SAC and DrQ-v2 policies. Experimental results show consistent improvements across multiple RL algorithms, neuron models, and simulation lengths.

**Compliance With Llm Reviewing Policy:**

Affirmed.

**Key Questions For Authors:**

See Weaknesses.

**Limitations:**

Yes.

**Strengths And Weaknesses:**

Strengths:
1: The paper shows an underexplored and important limitation of ANN-to-SNN conversion in continuous control settings.
2: The error analysis is insightful. The authors provide a clear investigation of the source of performance degradation and identify cross-step action error correlations as a key contributing factor.
3: The proposed CRPI is simple and well-motivated. It is easy to integrate into existing conversion frameworks.
4: The empirical evaluation is thorough. The method is tested on both vector-based and visual control tasks, across multiple RL algorithms, different spiking neuron models, and various simulation horizons.

Weaknesses:
1: Table 4 reports energy consumption only for three neurons under specific simulation horizons. The authors should provide more comprehensive energy analyses across more tasks, neuron models, and simulation settings.
2: In the MuJoCo and DMC experiments, the maximum episode length is limited to 1000 interaction steps. It would be useful to evaluate the method on longer-horizon control tasks to better understand the stability and long-term effects of cross-step residual propagation.
3: Minor typographical and grammatical errors such as “Equitation” in line 308 and “channelly” in line 302.

---

> ### Author Rebuttal · Authors · 2026-03-30
>
> Thank you for your time and effort invested in reviewing our paper. We will address all the questions you have raised.
> > W1: More comprehensive energy analysis
>
> Thank you for this valuable suggestion. We have extended the energy consumption analysis and reported the results in Table R1, following the same evaluation protocol as in Table 4. This covers a broader range of spiking neuron models, simulation steps, and different tasks in the DeepMind Control suite.
>
> Table R1: Energy consumption of SNNs in the DeepMind Control suite (μJ).
> | Module | T | Acrobot Swingup | Cartpole Swingup|Cheetah Run|Finger Spin|Quadruped Walk|Reacher Easy|Average|
> |:-:|:-:|:-:|:-:|:-:|:-:|:-:|:-:|:-:|
> |ANN|-|566.8|566.8|566.8|566.8|566.9|566.8|566.8|
> |IF|32|14.44|14.59|17.57|17.25|16.68|17.57|16.35|
> |IF|64|28.85|28.99|35.04|34.47|33.25|35.00|32.60|
> |SNM|8|3.85|4.23|5.16|4.83|5.08|4.99|4.69|
> |SNM|16|7.77|8.73|10.69|9.81|10.32|10.10|9.57|
> |MT|2|1.63|1.78|2.47|2.19|2.29|2.17|2.09|
> |DC|2|1.08|1.28|1.79|1.59|1.66|1.62|1.50|
>
> The results consistently show that:
> 1. All converted SNNs consume substantially less energy than their ANN counterparts;
> 2. Fewer simulation steps generally lead to higher energy efficiency;
> 3. SNNs with Differential Coding (DC) achieve the lowest energy consumption across all settings.
>
> These results further validate the energy efficiency of the converted SNNs.
>
>
> > W2: Evaluations on longer-horizon control tasks
>
> Thank you for this insightful question. We have further extended the maximum interaction horizon to 10,000 steps in the MuJoCo environments to evaluate long-term stability. Tables R2 and R3 report the performance of converted SNNs trained using DDPG and TD3, with the Integrate-and-Fire (IF) neurons.
>
> Table R2: Long-horizon performance (conversion of DDPG agents).
> |Environment|ANN|T=8 without CRPI|T=8 with CRPI|T=16 without CRPI|T=16 with CRPI|T=32 without CRPI|T=32 with CRPI|
> |:-:|:-:|:-:|:-:|:-:|:-:|:-:|:-:|
> |HalfCheetah-v4|56208|11874|**14251**|19971|**25245**|27271|**36685**|
> |Hopper-v4|5120|1097|**1410**|1770|**2123**|2422|**3805**|
> |Walker2d-v4|2514|3340|**4263**|2851|**4326**|3062|**3292**|
> |Average Performance Ratio|100.00%|58.46%|**74.13%**|61.16%|**86.14%**|72.54%|**90.16%**|
>
> Table R3: Long-horizon performance (conversion of TD3 agents).
> |Environment|ANN|T=8 without CRPI|T=8 with CRPI|T=16 without CRPI|T=16 with CRPI|T=32 without CRPI|T=32 with CRPI|
> |:-:|:-:|:-:|:-:|:-:|:-:|:-:|:-:|
> |Ant-v4|51519|5327|**11921**|29722|**40358**|45226|**57459**|
> |HalfCheetah-v4|129282|12864|**19904**|24655|**32112**|52411|**59222**|
> |Hopper-v4|36879|36547|**36677**|36826|**36830**|36818|**36843**|
> |Walker2d-v4|50683|5474|**40353**|33508|**49070**|44675|**50377**|
> |Average Performance Ratio|100.00%|32.55%|**55.40%**|60.68%|**74.96%**|78.08%|**89.16%**|
>
> The results show that vanilla SNNs suffer from significant performance degradation in longer-horizon settings, which we attribute to the accumulation of errors over decision steps. In contrast, our CRPI mechanism consistently improves performance and narrows the gap to ANN baselines across all settings and environments.
>
>
> > W3: Minor presentational errors
>
> Thank you for the careful reading. We have corrected all these errors and have carefully checked the manuscript to ensure overall clarity.

---

> > ### Author Rebuttal · Reviewer_UPqJ · 2026-04-03
> >
> > thanks for the author's detailed extended work, all my concerns have been addressed.

---

> > > ### Author Response · Authors · 2026-04-07
> > >
> > > We sincerely appreciate your recognition of our work. We are pleased that our rebuttal has resolved your concerns.

---

### Official Review · Reviewer_snnE · 2026-03-13

**Soundness:** 2
**Presentation:** 3
**Significance:** 2
**Originality:** 3
**Overall Recommendation:** 6
**Confidence:** 4

**Summary:**

The paper investigates why training-free ANN-to-SNN conversion suffers more severe performance degradation in continuous control reinforcement learning compared with classification or discrete control tasks. It attributes the key factor driving the amplification of state distribution shift to the cross-step temporal correlation of action errors. To address this issue, the paper proposes Cross-Step Residual Potential Initialization (CRPI), a simple training-free mechanism that carries residual membrane potentials across decision steps to decorrelate errors. This approach significantly restores performance across multiple conversion schemes on the MuJoCo and DeepMind Control Suite benchmarks.

**Compliance With Llm Reviewing Policy:**

Affirmed.

**Key Questions For Authors:**

see above

**Limitations:**

see above

**Strengths And Weaknesses:**

Strengths:

1.The paper identifies and empirically validates a clear mechanism—positive temporal correlation of action errors across decision steps—to explain why conversion errors are amplified in continuous control tasks.

2.In conventional ANN-to-SNN reinforcement learning conversion, the network state is reset at every environment step. Instead of performing residual membrane potential calibration only within a single decision window, this work introduces cross-step state preservation, which is compatible with multiple neuron models and conversion pipelines.

Weaknesses:

1.The proposed method introduces the residual membrane potential from the previous execution step into the next step. This operation can be interpreted as retaining partial state memory across decision steps. While this is acceptable for control tasks, it strictly changes the policy class relative to a stateless ANN baseline. I therefore recommend an ablation to clarify whether the performance gains come from “more accurate ANN→SNN conversion” or from “introducing cross-step memory.” The authors should compare against SNNs that explicitly introduce memory via training (e.g., recurrent/spiking architectures), for example GRSN [1].
[1] Qin, Lang, et al. “GRSN: Gated recurrent spiking neurons for POMDPs and MARL.” Proceedings of the AAAI Conference on Artificial Intelligence, Vol. 39, No. 2, 2025.

2.The theoretical derivation relies on several assumptions (for example, that the final membrane potential is weakly dependent on its initialization) which may not hold universally, making the method somewhat heuristic in nature despite its practical effectiveness. Line 1273 mentions “and weakly dependent on its initialization”, the authors should provide further discussion or cite supporting literature for this claim.

3.The hyperparameter α is selected via grid search using environment returns, but the paper does not state how many episodes were used—does this constitute significant extra environment interactions? Practically, how should α be chosen without incurring large additional environment interaction? In Tables 2 and 3, different values of α are chosen for different tasks. It would be useful to evaluate robustness by tuning α on a subset of validation tasks and then fixing it for unseen tasks.
4.The authors already compare their method with the directly trained approach Spiking WM in Table 3. An additional question is whether α could be learned through direct training, and how such a learned parameter would compare with the grid-search-based setting.

---

> ### Author Rebuttal · Authors · 2026-03-30
>
> We appreciate the time and effort you invested in reviewing our paper. We would like to address your concerns as follows.
> > W1: Where the performance gains come from
>
> Thank you for this insightful suggestion. We agree that introducing residual membrane potentials across steps can be interpreted as incorporating cross-step memory. We will include related work such as GRSN [1] in the revision:
>
> `GRSN introduces gated recurrent mechanisms and demonstrates strong performance on partially observable tasks. This line of work is complementary to ours, as it enhances SNN architectures via training, whereas our method improves ANN-to-SNN conversion precision.`
>
> To study the source of performance gains, we integrate CRPI into a directly trained SNN and evaluate its effect. The SNN is trained using TD3 for 1M environment steps with T=5 simulation steps, following the standard training method in [2].
>
> Table R1: Effect of cross-step memory on directly trained SNNs.
> |Environments|Vanilla SNN|SNN with cross-step memory|Performance gain|
> |:-:|:-:|:-:|:-:|
> |Ant-v4|4275|4416|3.29%|
> |HalfCheetah-v4|9683|9686|0.03%|
> |Hopper-v4|2956|2974|0.63%|
> |Walker2d-v4|4332|4332|0.00%|
>
> Table R1 shows that adding cross-step memory yields only marginal improvements (<1% on average). This is negligible compared to the gains from CRPI, indicating that the primary benefit comes from improved ANN-to-SNN conversion accuracy (by reducing error correlations) rather than cross-step memory.
> > W2: On the assumption that "the final membrane potential is approximately uniformly distributed in (0, θ) and weakly dependent on its initialization"
>
> Thank you for this insightful question. To empirically validate this assumption, we analyze the distribution of final membrane potentials for a converted TD3 agent (IF neuron, T=64) in the Hopper-v4 environment.
>
> Table R2: Distribution of final membrane potentials under different initializations.
> |Initialization|($-\infty$, 0]|(0, 0.1θ]|(0.1θ, 0.2θ]|(0.2θ, 0.3θ]|(0.3θ, 0.4θ]|(0.4θ, 0.5θ]|(0.5θ, 0.6θ]|(0.6θ, 0.7θ]|(0.7θ, 0.8θ]|(0.8θ, 0.9θ]|(0.9θ, θ]|(θ, $+\infty$)|
> |:-:|:-:|:-:|:-:|:-:|:-:|:-:|:-:|:-:|:-:|:-:|:-:|:-:|
> |0|85.3%|1.4%|1.5%|1.5%|1.5%|1.5%|1.5%|1.4%|1.5%|1.4%|1.4%|0.2%|
> |0.25θ|85.2%|1.5%|1.5%|1.5%|1.5%|1.5%|1.4%|1.4%|1.5%|1.4%|1.4%|0.2%|
> |0.5θ|85.2%|1.5%|1.5%|1.5%|1.5%|1.5%|1.4%|1.4%|1.4%|1.4%|1.4%|0.2%|
> |0.75θ|85.1%|1.5%|1.5%|1.5%|1.5%|1.5%|1.5%|1.5%|1.4%|1.4%|1.4%|0.2%|
> |θ|85.1%|1.5%|1.5%|1.5%|1.5%|1.5%|1.5%|1.5%|1.5%|1.4%|1.4%|0.2%|
>
> Table R2 shows that most neurons have negative potentials, where they are mostly inactive (and clipped in CRPI), and are thus irrelevant to the mechanism. For active neurons, membrane potentials are approximately uniformly distributed over (0, θ], with only a small fraction exceeding θ (and are also mostly clipped). Furthermore, this distribution remains nearly unchanged across initializations, indicating weak dependence on initialization.
>
> We will include this empirical evidence and discussion in the revised manuscript to better justify the assumption.
> > W3: Selecting hyperparameter α
>
> The grid search evaluates 11 candidate α values over 10 trials each, but these additional interactions are not strictly required. In practice, setting α in [0.2, 0.4] already yields strong and stable performance without additional environment interactions.
>
> To further evaluate the robustness, we fix α to a constant value and report average performance across 6 tasks in the DeepMind Control Suite (using IF neuron).
>
> Table R3: Relative performance of CRPI with fixed α.
> |T|Vanilla SNN|CRPI with α=0.2|CRPI with α=0.3|CRPI with α=0.4|CRPI with environment-specific α|
> |:-:|:-:|:-:|:-:|:-:|:-:|
> |32|40.77%|70.61%|69.25%|68.39%|74.19%|
> |64|69.59%|90.02%|90.06%|89.53%|91.84%|
>
> Table R3 shows that:
> 1. CRPI consistently improves over vanilla SNNs across different α values;
> 2. The performance is relatively insensitive to the exact choice of α in [0.2, 0.4];
> 3. Task-specific tuning of α brings only marginal gains.
>
> These results indicate that α can be fixed and directly transferred to unseen tasks without additional interaction or tuning.
> > W4: Whether α could be learned through direct training
>
> Thank you for this insightful suggestion. At this stage, we have not developed a mechanism to learn α through direct optimization.
>
> Our primary objective in this work is to identify key challenges in ANN-to-SNN conversion for continuous control and propose a simple yet effective solution. We therefore adopt grid search to select α to evaluate its effect while keeping the method simple. Learning α through gradient-based optimization or adaptive schemes is a promising direction, which we leave for future work.
>
> References:
>
> [1] Qin, L., et al. (2025). GRSN: Gated recurrent spiking neurons for POMDPs and MARL. AAAI.
>
> [2] Tang, G., et al. (2021). Deep reinforcement learning with population-coded spiking neural network for continuous control. CoRL.

---

> > ### Author Rebuttal · Reviewer_snnE · 2026-04-01
> >
> > My concerns are well addressed.

---

> > > ### Author Response · Authors · 2026-04-02
> > >
> > > We are encouraged that your concerns have been addressed. Thank you very much for your time and for your positive assessment of our work!

---

### Official Review · Reviewer_F8Wu · 2026-03-15

**Soundness:** 3
**Presentation:** 3
**Significance:** 3
**Originality:** 3
**Overall Recommendation:** 4
**Confidence:** 3

**Summary:**

I like the storyline of this paper, which analyzes the conversion error step by step and proposes a CRPI method that corrects the direction of accumulating conversion errors. The theory looks well explained to me and is easy to follow. The experiments also seem solid. My main concern lies in the explanation of energy consumption.

**Compliance With Llm Reviewing Policy:**

Affirmed.

**Final Justification:**

I recommend accept.

**Key Questions For Authors:**

See weakness

**Limitations:**

yes

**Strengths And Weaknesses:**

Strengths:

An easy-to-follow storyline. This paper proposes a simple but useful CRPI method. Also, this paper explores SNNs for continuous control, a domain that lacks strong SNN baselines.

Weakness:
1. The cost of CRPI should be discussed. Equation 22 introduces additional residual potential computation, and it also needs storing the membrane potential from previous steps, and scalar multiplication by $\alpha$. Has this energy cost been discussed and included in the “77 fJ” per SOP? Also, what is the energy overhead percentage of the CRPI method?

2. Usually, weight quantization is adopted on energy-constrained platforms. If both ANNs and SNNs use low-bit weights (e.g., 4-bit), the energy gap between low-bit MAC and ACC may become much smaller. It's better to clarify whether the claimed energy advantage of SNNs still holds under such low-bit deployment settings since authors declare their application on resource-constrained control systems such as robotics and embedded platforms. In addition, SNNs may also suffer from extra weight access and data movement overhead. Could the authors discuss these hardware-related energy overheads in comparison with ANNs? The energy analysis would be strengthened if the authors could include the above discussion.

A minor question: In Section 4.2, the authors state that the error comes from ANN-to-SNN conversion. If so, why not directly train an SNN instead? Does a directly trained SNN suffer from the same problem as converted SNNs?

---

> ### Author Rebuttal · Authors · 2026-03-30
>
> We would like to thank you for your constructive questions and suggestions. We are grateful for the opportunity to address the points you raised.
> > W1: The cost of CRPI
>
> Thank you for this insightful question. CRPI introduces negligible overhead (<1%) and is thus not included in Table 4.
>
> **The storage of membrane potentials from previous steps:** In typical neuromorphic hardware, membrane potentials are maintained by local capacitive elements within each neuron [1]. Therefore, storing and accessing the residual membrane potential does not require explicit buffering or off-chip memory access and introduces negligible hardware overhead.
>
> **The computational overhead of CRPI:** During deployment, the parameter $\alpha$ in CRPI can be selected from the set {0, 0.125, 0.25, 0.375, 0.5, 0.625, 0.75, 0.875, 1}, where each value can be expressed as a sum of powers of 1/2. The multiplication between $\alpha$ and the residual membrane potential can be efficiently implemented using at most three bitwise shifts and three floating-point accumulation operations (ACs). Other operations in CRPI (e.g., addition and clipping) do not involve multiplication. This design effectively eliminates multiplication operations during deployment, leading to minimal overhead.
>
>
> Table R1: Average computational overhead of CRPI on the DeepMind Control Suite.
> Spiking Neuron|T|ACs in CRPI|ACs in Forward Propagation|CRPI overhead|
> |:-:|:-:|:-:|:-:|:-:|
> |IF|32|$1.76\times10^5$|$2.12\times10^8$|0.08%|
> |MT|2|$1.88\times10^5$|$2.71\times10^7$|0.69%|
>
> Table R1 reports the average number of ACs introduced by CRPI compared to standard forward propagation. Across different neuron models, CRPI contributes less than 1% additional ACs, confirming its negligible overhead.
>
> > W2(a): Compared with quantization approaches
>
> Quantization is also an effective approach for reducing energy consumption on resource-constrained platforms. To provide a fair comparison, we consider a low-bit setting where both weights and activations are quantized.
>
> Following prior estimates, we adopt energy costs of 0.03 pJ per 8-bit integer addition and 0.2 pJ per multiplication as reported in [2]. For lower bit-widths, we assume $\mathcal{O}(N)$ complexity for addition and $\mathcal{O}(N^2)$ complexity for multiplication. Under this assumption, 4-bit addition and multiplication consume approximately 0.015 pJ and 0.05 pJ, respectively.
>
> Table R2: Inference-time energy consumption under quantization.
> ||ANN|IF (T=16)|MT (T=2)|
> |-|:-:|:-:|:-:|
> |Addition|$2.27\times10^7$|$2.12\times10^8$|$2.71\times10^7$|
> |Multiplication|$2.27\times10^7$|0|0|
> |Energy (Int8)|5.21 μJ|6.37 μJ|**0.81 μJ**|
> |Energy (Int4)|1.25 μJ|3.18 μJ|**0.41 μJ**|
>
> Table R2 reports the average inference-time energy consumption of ANNs and converted SNNs on the DeepMind Control Suite under different quantization settings. Quantization reduces energy consumption for both ANNs and SNNs. Under low-bit settings, quantized ANNs consume less energy than SNNs with vanilla IF neurons. However, when using more advanced Multi-Threshold (MT) neurons, SNNs remain more energy-efficient than quantized ANNs. The energy advantage of SNNs still holds under low-bit deployment, and SNNs remain a competitive solution for resource-constrained control systems.
>
> > W2(b): Extra weight access and data movement overhead
>
> Thank you for this insightful question. We agree that on conventional von Neumann architectures (e.g., GPUs and TPUs), SNNs may incur extra memory access overhead.
>
> However, most SNN-friendly hardware platforms adopt in-memory computing (IMC) architectures [3]. By performing computations directly within memory arrays where the weights are stored, IMC eliminates the need for weight access across global buses. Moreover, with specialized peripheral circuit designs, operations like membrane potential accumulation can be executed without additional buffering [1]. These designs significantly reduce weight access and data movement overhead, enabling SNNs to achieve their theoretical efficiency advantages.
>
> > Q: Why not directly train an SNN instead?
>
> Directly trained SNNs indeed do not suffer from "conversion error". However, in challenging vision-based continuous control tasks (such as those in the DeepMind Control Suite), their performance remains relatively limited and falls significantly behind conversion approaches (as shown in Table 3).
>
> Moreover, conversion methods offer a practical advantage in RL settings. By reusing pretrained ANN policies, they substantially reduce the need for costly environment interactions required for training from scratch.
>
> References:
>
> [1] Indiveri, G., et al. (2015). Neuromorphic architectures for spiking deep neural networks. IEDM.
>
> [2] Horowitz, M. (2014). Computing’s energy problem (and what we can do about it). ISSCC.
>
> [3] Montuschi, P., et al. (2023). In-memory computing: the emerging computing topic in the post-von Neumann era. Computer.

---

> > ### Author Rebuttal · Reviewer_F8Wu · 2026-04-01
> >
> > One more thing that needs to be noticed is that if you use shifting to do the simplification, the membrane potential had better be an integer. Also, make sure the accuracy loss caused by shifting is fully considered (e.g., 3 >> 1 = 1).

---

> > > ### Author Response · Authors · 2026-04-01
> > >
> > > We sincerely thank you for helping us to make the hardware analysis more rigorous. We will carefully keep these important points in mind and include a detailed discussion on this in the camera-ready version and our future work.

---

### Decision · Program_Chairs · 2026-04-30

**Decision:**

Accept (regular)

**Comment:**

Summary of reviews. The reviewers are overall positive about this submission. The paper studies a relatively underexplored but important setting: ANN-to-SNN conversion for continuous control reinforcement learning. The central claim is that small conversion-induced action errors can become temporally correlated across decision steps, leading to cumulative state-distribution shift and amplified performance degradation. To address this issue, the paper proposes Cross-Step Residual Potential Initialization (CRPI), a simple gradient-free mechanism that carries residual membrane potentials across decision steps to reduce temporally correlated conversion errors. The method is evaluated on MuJoCo and DeepMind Control Suite tasks, across multiple RL algorithms, neuron models, and simulation horizons.

The final reviewer scores are positive: one Weak Accept, one Accept, and two Strong Accepts. All four reviewers ultimately recommend acceptance after the rebuttal.

Key strengths.

Important and underexplored problem: ANN-to-SNN conversion in continuous control.
Clear diagnosis of error amplification via temporally correlated action errors.
Simple and practical method, CRPI, compatible with existing conversion pipelines.
Broad empirical validation across control benchmarks, RL algorithms, and neuron models.
Rebuttal added useful evidence on robustness, overhead, energy, and long-horizon performance.
Key weaknesses and rebuttal assessment. *Some theoretical assumptions were initially under-justified, but the rebuttal added empirical support. *The “training-free” framing was overstated; “gradient-free” is more accurate. *Alpha tuning requires some environment interaction, though fixed-alpha results reduce this concern. *Initial energy and variability analyses were incomplete, but substantially improved in rebuttal. *Several important experiments should be incorporated into the final paper.

Discussion outcome.

All reviewers marked their concerns as fully resolved after the rebuttal. Reviewer F8Wu maintained a Weak Accept and recommended acceptance. Reviewer UPqJ confirmed that the extended energy and long-horizon analyses addressed their concerns. Reviewer snnE stated that their concerns were well addressed. Reviewer 66vP raised their score after the rebuttal, explicitly noting that the responses on variability, alpha tuning, fine-tuning/distillation, and the revised “gradient-free” framing were satisfactory.

Overall, this is a solid and timely contribution. The paper’s main value lies not only in the proposed CRPI mechanism, but also in clearly diagnosing why ANN-to-SNN conversion is more fragile in continuous-control settings than in classification or discrete-control settings. I recommend acceptance, with the expectation that the authors incorporate the rebuttal analyses and clarify the remaining methodological assumptions in the camera-ready.

Recommendation and Normination:

I recommend this paper for acceptance given the consistent positive feedback.

I am not actively nominating this paper for Oral/SpotLight. While the paper is a solid and timely contribution with positive reviewer consensus after rebuttal, its impact appears strongest within the ANN-to-SNN conversion and neuromorphic RL subcommunity rather than broadly across ICML. Several important clarifications and additional experiments were also added only during rebuttal. I therefore recommend acceptance but do not view it as a strong oral or spotlight candidate.